# Latent Retrieval Augmented Generation of Cross-Domain Protein Binders

**Zishen Zhang**[1,2] **Xiangzhe Kong**[1,2] **Wenbing Huang**[3*] **Yang Liu**[1,2*]
[1]Dept. of Comp. Sci. & Tech., Tsinghua University
[2]Institute for AIR, Tsinghua University
[3]Gaoling School of Artificial Intelligence, Renmin University of China

## Abstract

Designing protein binders targeting specific sites, which requires to generate realistic and functional interaction patterns, is a fundamental challenge in drug discovery. Current structure-based generative models are limited in generating nterfaces with sufficient rationality and interpretability. In this paper, we propose **R**etrieval-**A**ugmented **Di**ffusion for **A**lig**n**ed interfa**ce**(**RADiAnce**), a new framework that leverages known interfaces to guide the design of novel binders. By unifying retrieval and generation in a shared contrastive latent space, our model efficiently identifies relevant interfaces for a given binding site and seamlessly integrates them through a conditional latent diffusion generator, enabling cross-domain interface transfer. Extensive exeriments show that **RADiAnce** significantly outperforms baseline models across multiple metrics, including binding affinity and recovery of geometries and interactions. Additional experimental results validate cross-domain generalization, demonstrating that retrieving interfaces from diverse domains, such as peptides, antibodies, and protein fragments, enhances the generation performance of binders for other domains. Our work establishes a new paradigm for protein binder design that successfully bridges retrieval-based knowledge and generative AI, opening new possibilities for drug discovery.

## 1 Introduction

Designing binders that target specific sites on proteins is a fundamental problem with broad implications in drug discovery, structural biology, and beyond [54, 45]. The core challenge involves engineering molecular interfaces that exhibit desired interaction patterns with a target surface, such as hydrogen bonding, hydrophobic packing, and $\pi-\pi$ stacking [45]. Traditional approaches rely on sampling and optimizing within physical or statistical energy landscapes [3, 8, 22], while recent advances have shifted towards direct structure-based generation of binders with deep generative models, such as diffusion and flow matching [53, 40].

Despite these advances, current methods remain limited in rationality and interpretability due to their inability to leverage prior knowledge from existing structurally similar interfaces. Most approaches conduct generation solely conditional on the target binding site [30, 40], ignoring the wealth of reusable interaction patterns present in existing protein complexes. In contrast, it is routine for human experts (e.g. medicinal chemists or structural biologists) to rely on similar known interfaces to guide rational design [48, 17]. We thus argue that a better generative paradigm should identify and incorporate analogous interfaces from existing databases to improve both rationality and interpretability. However, incorporating interface retrieval into the generation process poses several challenges. First, the retriever should identify relevant interfaces from a large database using only the binding site as input. Existing retrieval methods typically assume a known binder structure (e.g., for inverse

---

*Correspondence to Wenbing Huang <hwenbing@126.com>, Yang Liu <liuyang2011@tsinghua.edu.cn>

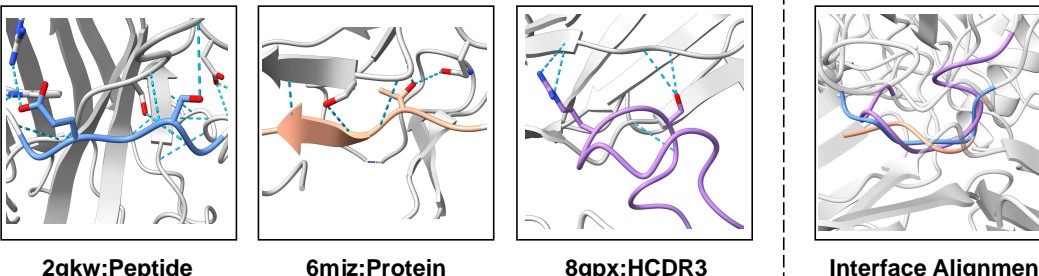

| **2gkw:Peptide** | **6mjz:Protein** | **8gpx:HCDR3** | **Interface Alignment** |

Figure 1: Visualization of interface similarity across antibodies, proteins, and peptides, highlighting similar interaction patterns among diverse binder types.

folding [51]), which is inapplicable in our setting. Second, while binders may vary in form (e.g., cyclic peptides, linear peptides, antibodies) due to different biochemical requirements, the underlying interaction motifs are often generalizable across binders of different types and domains, as illustrated in Fig. 1. Therefore, the similarity metric used in the retriever should capture shared interactions between cross-domain interfaces and binding sites. Moreover, a mechanism is needed to integrate the retrieved information into the generative process, effectively focusing on relevant parts for different generated regions.

To address the challenges, this work introduces **R**etrieval-**A**ugmented **Di**ffusion for **Align**ed in­terfa**ce**(**RADiAnce**), a novel framework that unifies retrieval and generation in a shared contrastive latent space. We begin by constructing a comprehensive interface database spanning peptides [30], antibodies [14], and protein fragments [30], where each entry comprising a binding site and a corre­sponding ligand interface. We then train an all-atom variational autoencoder (VAE) [31] augmented with contrastive learning to derive an interaction-aligned and retrieval-friendly latent space. The encoder maps each binding site and interface independently to separate latent vectors, which are supervised by a contrastive loss to align for positive pairs and repel for negative ones. At inference time, the encoder is used to map database entries of interface pairs into key-value vectors, enabling fast and accurate retrieval using simple dot-product similarity. A diffusion model is then trained in the same latent space, conditioned on the retrieved interface embeddings. These embeddings act as compressed yet informative prompts, integrated through a combination of cross-attention and residual MLPs [50]. This enables the model to leverage relevant priors (such as residue preferences, contact geometries, and binding orientations) from the retrieved examples, resulting in rational, interpretable binder generations with physically plausible interfaces.

We summarize our contributions as follows:

- **Contrastive Latent Space for Cross-Domain Interface Retrieval**. We propose an all-atom VAE that is trained contrastively to align latent representations between binding sites and their interfaces. This latent space enables accurate and a unified similarity metric across diverse binder types such as antibodies, peptides, and protein fragments.

- **Retrieval-Augmented Latent Diffusion**.To the best of our knowledge, we are the first to apply retrieval-augmented generation for the sequence-structure codesign of protein binders. We introduce a latent diffusion model that operates in the same latent space as the retriever, seamlessly incorporating retrieved interface embeddings via cross-attention and residual conditioning to guide generation.

- **Extensive Benchmarking and Analysis**. We benchmark **RADiAnce** on peptide and antibody design tasks, demonstrating significant improvements over strong baselines in recovering sequence, structure, and interaction patterns. Further analysis confirms that the gains stem from the ability of our model in mimicing retrieved interaction motifs. We also show that leveraging interfaces from other binder types (e.g., antibodies to aid peptide generation) improves performance, veryfing the rationality of cross-domain transfer.

## 2 Related Work

**Peptide Design**    Peptide design has transitioned from classical energy-based sampling [8, 22] to deep generative modeling [35]. Recent approaches, such as PepFlow [36] and PPFlow [39], employ multi-

modal flow matching to jointly model residue types, backbone orientations, $C_\alpha$ positions, and side-chain dihedrals, setting new benchmarks in peptide generation. PepGLAD [30] advances this direction by applying geometric latent diffusion models for sequence–structure co-design. Parallel efforts using language models, including PeptideGPT [47], generate functional peptides from sequence space and filter candidates through structure prediction tools like AlphaFold [27] to ensure 3D viability.

**Antibody Design** Antibody design has similarly evolved from traditional physical sampling [3] to deep learning frameworks [46, 4, 41]. Early methods focused on CDR loop generation without antigen context, while later works such as MEAN [28] and DiffAb [40] introduced antigen-conditioned generation. Diffusion-based models like AbDiffuser [41] and GeoAB [38] further improve structure-aware antibody design by incorporating physical constraints, while HERN [26] captures long-range dependencies via hierarchical graph decoding. Additionally, language-model-based approaches have gained prominence, PALM-H3 [19] and MAGE [52] adopt encoder-decoder architectures to generate CDR sequences or full variable regions conditioned on antigen features.

**Retrieval-Augmented Generation(RAG)** RAG integrates external knowledge into generative models to enhance fidelity, interpretability, and controllability. Initially developed for NLP tasks like question answering [34, 18], RAG frameworks pair a retriever with a generator to access dynamic knowledge during inference. In vision, retrieval-conditioned diffusion models [10, 11] improve image synthesis by incorporating structure-consistent references. Recently, RAG has emerged as a powerful paradigm for targeted molecular design. For instance, f-RAG introduces a dynamic fragment vocabulary to improve small-molecule generation via fragment-level retrieval [55]. Structure-based approaches such as IRDiff leverage high-affinity ligands as retrieval references to guide diffusion-based 3D molecular generation [32], while READ incorporates pocket-matched scaffold embeddings into SE(3)-equivariant diffusion to ensure both structural validity and affinity optimization [23]. RADAb further explored retrieval for antibody design but was limited to sequence inverse folding with fixed structures [51].

Despite these advances, existing methods do not address the challenge of protein binder co-design, which requires simultaneous modeling of sequence and structure, nor do they provide a unified retrieval framework across modalities such as peptides and antibodies. Our method, **RADiAnce**, bridges this gap by learning a contrastive latent space that aligns interaction interfaces across diverse binder modalities and by introducing a retrieval-augmented latent diffusion model that enables cross-domain motif retrieval to guide joint sequence–structure design.

## 3 Method

We begin the introduction of our method by establishing the notations and definitions (§3.1). Our framework consists of a contrastive atomic variational autoencoder (§3.2) to define a compressed latent space for target-based retrieval, and a conditional latent diffusion model defined on the same latent space, which learns from the retrieved interfaces (§3.3).

### 3.1 Notations and Definitions

We represent a molecular complex as a pair of graphs $(\mathcal{X}, \mathcal{Y})$, where $\mathcal{X}$ denotes the binder molecule and $\mathcal{Y}$ represents the binding site on the target protein. Each graph, either $\mathcal{X}$ or $\mathcal{Y}$, comprises a node set $\mathcal{V} = \{(\boldsymbol{A}_i, \vec{\boldsymbol{X}}_i)\}$ and an edge set $\mathcal{E} = \{(\boldsymbol{B}_i, \boldsymbol{B}_{ij}) \mid i \neq j\}$, where $\boldsymbol{A}_i \in \mathbb{Z}^{n_i}$ and $\vec{\boldsymbol{X}}_i \in \mathbb{R}^{n_i \times 3}$ are the element types and 3D coordinates of the $n_i$ atoms in block $i$, with $\boldsymbol{B}_i$ and $\boldsymbol{B}_{ij}$ denoting intra- and inter-block chemical bonds. We treat each standard amino acid as one block, and use principal subgraph decomposition algorithm to decompose other entities (e.g. non-standard amino acids) into blocks of frequently occurring chemical motifs. The resulting vocabulary $\mathbb{V}$ thus extends beyond standard amino acids and also includes principal subgraphs derived from non-standard components and we denote the type of block $i$ as $s_i$ [31]. The binding site $\mathcal{Y}$ is defined as residues within a 10Å distance cutoff from the binder, based on $C_\beta$ atoms. We construct a key-value database $\mathbb{D} = \{(\mathcal{Y}_k, \mathcal{X}_k) \mid k = 1, 2, 3, ...\}$, using the binding site $\mathcal{Y}$ as the key and the corresponding binder interface $\mathcal{X}$ as the value. Given a query binding site $\mathcal{Y}$, our framework first retrieves potential reference interfaces $\mathbb{T}(\mathcal{Y} \mid \mathbb{D}) = \{\mathcal{X}_k \mid (\mathcal{Y}_k, \mathcal{X}_k) \in \mathbb{D}, \mathcal{Y}_k \text{ similar to } \mathcal{Y}\}$ from the database and then generates a new binder by modeling the conditional distribution $p_\theta(\mathcal{X} \mid \mathcal{Y}, \mathbb{T}(\mathcal{Y} \mid \mathbb{D}))$.

## 3.2 Contrastive Latent Space

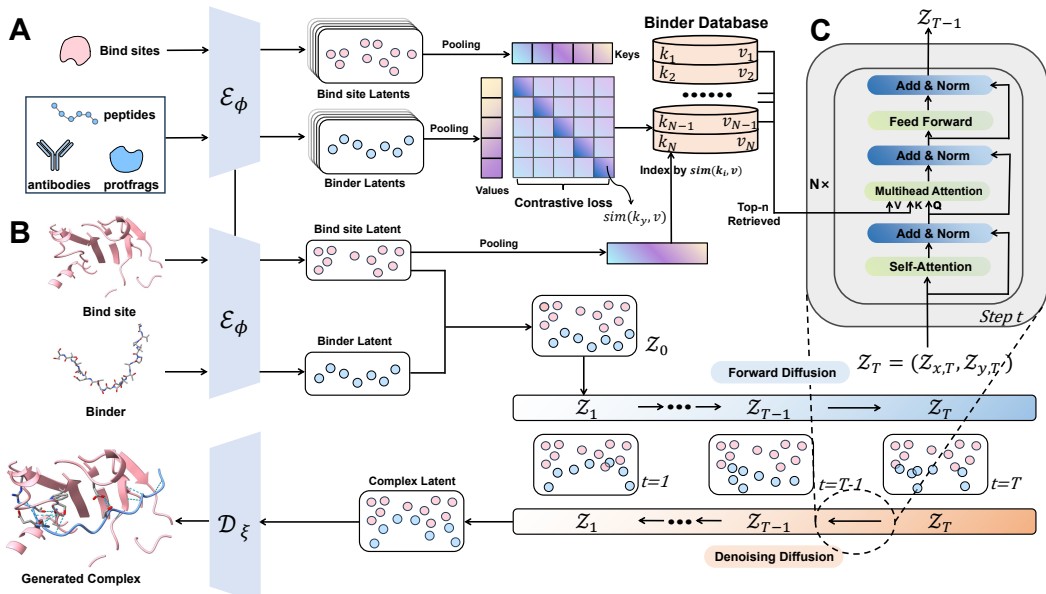

Figure 2: **Overview of RADiAnce.** (A) Cross-domain binding sites and binders are encoded into key/value latents and trained with a contrastive loss to derive a retrievable binder database. (B) Contrastive VAE aligns binding site and binder latents for accurate retrieval and conditional diffusion. (C) Conditional diffusion generator leverages the retrieved latents through cross-attention at every reverse step, progressively refining noisy features into sterically and chemically consistent complexes.

To enable reliable retrieval and informative conditioning during generation, we construct a compressed contrastive latent space using an atomic variational autoencoder [31], which is trained to support both interface retrieval and atom-level reconstruction. The encoder projects the binder $\mathcal{X}$ and binding site $\mathcal{Y}$ independently into two latent point clouds, $\mathcal{Z}_x$ and $\mathcal{Z}_y$, while the decoder reconstructs the binder $\mathcal{X}$ conditioned on $\mathcal{Y}$ and $\mathcal{Z}_y$:

$$\mathcal{Z}_x = \mathcal{E}_\phi(\mathcal{X}), \qquad \mathcal{Z}_y = \mathcal{E}_\phi(\mathcal{Y}), \qquad \hat{\mathcal{X}} = \mathcal{D}_\xi(\mathcal{Z}_x, \mathcal{Z}_y, \mathcal{Y}), \tag{1}$$

where latent variable $\mathcal{Z}_{x/y} = \{z_i, \vec{z}_i\}$ consists of a scalar embedding $z_i = W_z h_i \in \mathbb{R}^d$ derived from a trainable linear transformation of the hidden states $h_i$, and a 3D coordinate $\vec{z}_i \in \mathbb{R}^3$ for each block $i$. The decoder first predicts the block (i.e. residue) type from $z_i$ and then reconstructs atomic coordinates $\vec{X}_i$ via a flow matching process using a Gaussian prior centered at $\vec{z}_i$. The reconstruction loss includes cross-entropy (CE) over block types and mean squared error (MSE) over time-dependent vector fields:

$$\mathcal{L}_{\text{rec}}(i) = \text{CE}(p(\hat{s}_i), p(s_i)) + \mathbb{E}_{t \sim U(0,1)} \left[ \text{MSE}(\hat{\vec{V}}_i^t, \vec{V}_i^t) \right], \tag{2}$$

where $\hat{s}_i$ and $\hat{\vec{V}}_i^t \in \mathbb{R}^{n_i \times 3}$ denote the reconstructed block type and the vector field at time step $t$. Note that $z_i$ and $\vec{z}_i$ are sampled from the encoded distributions $\mathcal{N}(\mu_i, \sigma_i)$ and $\mathcal{N}(\vec{\mu}_i, \vec{\sigma}_i)$, respectively. To regularize the latent space, we apply Kullback–Leibler (KL) divergence to both scalar and coordinate embeddings [31] with prior distributions $\mathbb{N}(z_i; 0, I)$ and $\mathbb{N}(\vec{z}_i; \vec{c}_i, I)$, where $\vec{c}_i$ denotes the center of mass for block $i$:

$$\mathcal{L}_{KL}(i) = \lambda_1 \cdot D_{\text{KL}}(\mathcal{N}(0, I) \| \mathcal{N}(\mu_i, \text{diag}(\sigma_i))) + \lambda_2 \cdot D_{\text{KL}}(\mathcal{N}(\vec{c}_i, I) \| \mathcal{N}(\vec{\mu}_i, \text{diag}(\vec{\sigma}_i))), \tag{3}$$

where $\lambda_1$ and $\lambda_2$ define the strengths for constraining on the latent variables. To further endow the latent space with the ability to retrieve relevant interfaces given the binding site $\mathcal{Y}$, we implement contrastive training between the encoded embeddings of the binding site and the binder interfaces. Specifically, the graph-level embeddings of the binding site and the binder are derived by averaging the hidden states as $k = \frac{1}{\|\mathcal{Y}\|} \sum_{i \in \mathcal{Y}} h_i$, and $v = \frac{1}{\|\mathcal{X}\|} \sum_{i \in \mathcal{X}} h_i$, respectively. Subsequently, the

training objective maximize the similarity for authentic pairs $\{(\boldsymbol{k}_p, \boldsymbol{v}_q) \mid p = q\}$ and minimize it for mismatched pairs $\{(\boldsymbol{k}_p, \boldsymbol{v}_q \mid p \neq q\}$ in a double-sided manner [16]:

$$\mathcal{L}_{\text{retrieval}}(p) = -\log \frac{\exp(\langle \boldsymbol{k}_p, \boldsymbol{v}_p \rangle / \tau)}{\sum_{q=1}^{N} \exp(\langle \boldsymbol{k}_p, \boldsymbol{v}_q \rangle / \tau)} - \log \frac{\exp(\langle \boldsymbol{v}_p, \boldsymbol{k}_p \rangle / \tau)}{\sum_{q=1}^{N} \exp(\langle \boldsymbol{v}_p, \boldsymbol{k}_q \rangle / \tau)}, \quad (4)$$

where $\tau$, $N$, and $\langle \cdot, \cdot \rangle$ are the temperature, the batch size, and the standard Euclidean inner product. The temperature $\tau > 0$ controls the sharpness of the similarity distribution. The total training objective over the dataset $\mathcal{D}$ combines reconstruction, KL regularization, and contrastive retrieval:

$$\mathcal{L}(\mathcal{D}) = \sum_{p \in \mathcal{D}} (\sum_{i \in \mathcal{X}_p} \mathcal{L}_{\text{rec}}(i) + \sum_{i \in \mathcal{X}_p} \mathcal{L}_{\text{KL}}(i) + \mathcal{L}_{\text{retrieval}}(p)). \quad (5)$$

After training, the database $\mathbb{D}$ containing complexes of interest is processed by the contrastive encoder to derive key-value pairs $\mathbb{D}^{kv} = \{(\boldsymbol{k}_k, \boldsymbol{v}_k) \mid k = 1, 2, 3, ...\}$. During training and inference of the diffusion model, given an arbitrary binding site $\mathcal{Y}$, we use the encoder to obtain its key embedding $\boldsymbol{k}$, which is then used to query $\mathbb{D}^{kv}$ via inner product similarity. Additionally, to prevent trivial memorization and ensure generalization, we exclude the ground-truth ligand of the query complex from the retrieval database during this querying step. The top-$n$ nearest neighbors are retrieved based on this process and serve as template candidates for subsequent diffusion-based generation on $\mathcal{X}$, which are denoted as $\mathbb{T}^v = \{\boldsymbol{v}_k \mid k = 1, 2, ..., n\}$.

## 3.3 Retrieval-Conditioned Latent Diffusion

**Latent Diffusion**   Given that the variational autoencoder (VAE) encodes the binding site and binder into continuous latent representations, we directly define a diffusion process over these latent variables to model the conditional distribution $p(\mathcal{Z}_x | \mathcal{Z}_y, \mathbb{T}^v)$, where $\mathbb{T}^v$ contains the value vectors of the retrieved template interfaces. The forward process gradually corrupts the initial latent variables with Gaussian noise across a predefined number of steps $T$, transforming them into the standard Gaussian prior $\mathcal{N}(\boldsymbol{0}, \boldsymbol{I})$. Conversely, the reverse process denoises these samples step by step to recover the original latent vectors. Let $\vec{\boldsymbol{u}}_i^t = [\boldsymbol{z}_i^t, \vec{\boldsymbol{z}}_i^t]$ denote the state of block $i$ at timestep $t$, the forward process can be defined as follows [31]:

$$q(\vec{\boldsymbol{u}}_i^t \mid \vec{\boldsymbol{u}}_i^{t-1}) = \mathcal{N}(\vec{\boldsymbol{u}}_i^t; \sqrt{1 - \beta^t} \cdot \vec{\boldsymbol{u}}_i^{t-1}, \beta^t \boldsymbol{I}), \quad (6)$$

$$q(\vec{\boldsymbol{u}}_i^t \mid \vec{\boldsymbol{u}}_i^0) = \mathcal{N}(\vec{\boldsymbol{u}}_i^t; \sqrt{\bar{\alpha}^t} \cdot \vec{\boldsymbol{u}}_i^0, (1 - \bar{\alpha}^t)\boldsymbol{I}), \quad (7)$$

where $\bar{\alpha}^t = \prod_{s=1}^{s=t}(1 - \beta^s)$, and $\beta^t$ is the noise scale following the cosine schedule [43]. Therefore, state at time $t$ can be sampled given $\vec{\boldsymbol{u}}_i^0$ as $\vec{\boldsymbol{u}}_i^t = \sqrt{\bar{\alpha}^t}\vec{\boldsymbol{u}}_i^0 + \sqrt{1 - \bar{\alpha}^t}\boldsymbol{\epsilon}_i$, where $\boldsymbol{\epsilon}_i \sim \mathcal{N}(\boldsymbol{0}, \boldsymbol{I})$. The reverse process learns to denoise the latent samples [21] with retrieved templates as contexts:

$$p_\theta(\vec{\boldsymbol{u}}_i^{t-1} \mid \mathcal{Z}_x^t, \mathcal{Z}_y, \mathbb{T}^v) = \mathcal{N}(\vec{\boldsymbol{u}}_i^{t-1}; \vec{\boldsymbol{\mu}}_\theta(\mathcal{Z}_x^t, \mathcal{Z}_y, \mathbb{T}^v), \beta^t \boldsymbol{I}), \quad (8)$$

$$\vec{\boldsymbol{\mu}}_\theta(\mathcal{Z}_x^t, \mathcal{Z}_y, \mathbb{T}^v) = (\vec{\boldsymbol{u}}_i^t - \frac{\beta^t}{\sqrt{1 - \bar{\alpha}^t}} \boldsymbol{\epsilon}_\theta(\mathcal{Z}_x^t, \mathcal{Z}_y, \mathbb{T}^v, t)[i]) / \sqrt{\alpha^t}, \quad (9)$$

where $\alpha^t = 1 - \beta^t$. The parameterization of the denoising kernel $\boldsymbol{\epsilon}_\theta$ will be illustrated in the following paragraph. The training process miminizes the discrepancy between the predicted and true noise added during the forward process:

$$\mathcal{L}_{LDM} = \mathbb{E}_{t \sim \text{U}(1...T)}[\sum_i \|\boldsymbol{\epsilon}_i - \boldsymbol{\epsilon}_\theta(\mathcal{Z}_x^t, \mathcal{Z}_y, \mathbb{T}^v, t)[i]\|^2 / |\mathcal{Z}_x^t|], \quad (10)$$

where $|\mathcal{Z}_x^t|$ denotes the size of the latent point cloud.

**Template Integration**   The denoising kernel $\boldsymbol{\epsilon}_\theta(\mathcal{Z}_x^t, \mathcal{Z}_y, \mathbb{T}^v, t)$ is instantiated by an E(3)-equivariant transformer (EPT) [25] augmented with a cross-attention mechanism to incorporate information from the retrieved templates $\mathbb{T}^v$. This cross-attention module is placed between the self-attention and feed-forward layers in the original EPT architecture (Figure 2). Let $\boldsymbol{H} \in \mathbb{R}^{m \times h}$ denote the invariant hidden states of the $m$ latent nodes (blocks) with hidden dimension $h$, and let $\boldsymbol{T} \in \mathbb{R}^{n \times h}$ represent the embeddings of the retrieved top-$n$ templates. The queries, keys, and values are computed as:

$$\boldsymbol{Q} = \boldsymbol{H}\boldsymbol{W}^Q, \qquad \boldsymbol{K} = \boldsymbol{T}\boldsymbol{W}^K, \qquad \boldsymbol{V} = \boldsymbol{T}\boldsymbol{W}^V, \quad (11)$$

where $\boldsymbol{W}^Q, \boldsymbol{W}^K, \boldsymbol{W}^V \in \mathbb{R}^{h \times h}$ are learnable matrices for linear transformations. The hidden states are subsequently updated using a residual connection with the cross-attention output:

$$\boldsymbol{H}' = \text{LN}(\boldsymbol{H} + \text{Softmax}(\frac{\boldsymbol{Q}\boldsymbol{K}^\top}{\sqrt{h}})\boldsymbol{V}), \tag{12}$$

where LN denotes the layernorm [6]. This design enables the model to effectively integrate contextual information from the retrieved templates $\mathbb{T}$, guiding the refinement of latent variables toward mimicking key interaction patterns. We have also explored alternative conditioning mechanisms for incorporating template information, but these variants exhibit downgraded performance compared to the current cross-attention strategy, as demonstrated in Appendix D.

### 3.4 Inference Algorithm

During inference, the framework encodes the target binding-site graph $\mathcal{Y}$ into a key $\boldsymbol{k}_y$, retrieves the top-$K$ latent binders from the database via inner-product similarity to form the prompt set $\mathbb{T}^v$, and performs prompt-conditioned reverse diffusion from Gaussian noise to a clean latent $\hat{\mathcal{Z}}_0$, which is then decoded into the binder $\hat{\mathcal{X}}$. Owing to the modularity of retrieval and generation, the latent database $\mathbb{D}$ can be freely replaced at inference. The overall workflow is as follows:

---
**Algorithm 1** Inference Workflow

---
1: **Input**: Binding site structure $\mathcal{Y}$
2: $\boldsymbol{k}_y \leftarrow \mathcal{E}_\phi(\mathcal{Y})$                 {Encode binding site into query key}
3: **for** each $\boldsymbol{v}^{(j)} \in \mathbb{D}$ **do**
4:    $s_j \leftarrow \langle \boldsymbol{k}_y, \boldsymbol{v}^{(j)} \rangle$                 {Compute similarity by inner product}
5: **end for**
6: $\mathcal{I} \leftarrow \text{TopK}(\{s_j\})$                 {Indices of top-$K$ highest similarity scores}
7: $\mathbb{T}^v \leftarrow \{\boldsymbol{v}^{(k)} : k \in \mathcal{I}\}$                 {Retrieve binder latents to form prompt features}
8: $\mathcal{Z}_T \sim \mathcal{N}(0, \boldsymbol{I})$                 {Initialize latent variable from Gaussian}
9: **for** $t = T$ **down to** 1 **do**
10:    $\boldsymbol{\epsilon} \leftarrow \boldsymbol{\epsilon}_\theta(\mathcal{Z}_x^t, \mathcal{Z}_y, \mathbb{T}^v, t)$                 {Predict noise}
11:    $\mathcal{Z}_{t-1} \leftarrow \text{Denoise}(\mathcal{Z}_t, \boldsymbol{\epsilon})$                 {Refine latent variable}
12: **end for**
13: $\hat{\mathcal{X}} \leftarrow \mathcal{D}_\xi(\mathcal{Z}_0)$                 {Decode final latent to molecular graph}
14: **Output**: Generated binder structure $\hat{\mathcal{X}}$

---

## 4 Experiments

We evaluate **RADiAnce** from two key aspects: **Retrieval Reliability** (Section 4.1) and **Generation Performance** (Section 4.2), using peptide, antibody, and protein fragments from existing literature. For peptide design, we adopt PepBench [30], which includes 4,157 training and 114 validation complexes, with 93 test cases from the LNR benchmark [49]. For antibodies, we follow the literature [31] to use 9,473 training and 400 validation entries from SAbDab [14], and 60 test cases from the RAbD benchmark [3]. We further include ProtFrag [30], a dataset of 70,498 monomer-derived protein fragments, to assess the cross-domain referencing ability of our framework. To prevent data leakage, all retrieval operations are strictly limited to training data. Lastly, we demonstrate the potential of **RADiAnce** in real-world scenarios with a pipeline for de novo antibody design without relying on predefined frameworks (Section 4.3).

### 4.1 Retrieval Reliability

**Setup**  We first assess whether the latent embeddings produced by the variational encoder can reliably retrieve relevant interfaces, as accurate retrieval is essential for successful downstream generation. Although the model is trained on cross-domain data, retrieval evaluation is conducted using only the in-domain database (i.e., antibody or peptide) to create a more stringent testing scenario. Ground-truth binders are also included in the database for this setting, to evaluate whether the models can accurately retrieve the ground truths. We further perform ablation studies to assess

the effect of cross-domain training and the importance of jointly optimizing the VAE and contrastive objectives. As a baseline, we include random retrieval to illustrate the degree of structural and biological relevance achieved by our Contrastive VAE (**CVAE**).

**Metrics**   We evaluate retrieval reliability from two complementary perspectives: recall and precision. To assess recall, we compute the proportion of cases where the ground-truth binder ranks within the top $N\%$ of retrieved candidates (RC), denoted as **RC-N**%. For precision, we assess the similarity between the top-10 retrieved interfaces and the ground-truth binder in terms of shared interaction patterns (e.g., hydrogen bonds, hydrophobic contacts,etc) [2]. This is quantified by Interaction Type Overlap (**ITO**), which measures agreement between two distributions consisting of interaction types, with details provided in Appendix B.

Table 1: Results of recovery metrics for retrieval models. Columns "Pep", "Ab", and "ProtFrags" indicate whether peptide, antibody, and protein fragment data are used during training (✓ for included, ✗ for excluded). "All loss" = ✗ indicates using contrastive loss only.

| Model | Pep | Ab | ProtFrag | All loss | ITO(%) | RC-0.1% | RC-0.5% | RC-5% |
|---|---|---|---|---|---|---|---|---|
| **Antibody** | | | | | | | | |
| **Random** | ✗ | ✗ | ✗ | ✓ | 17.71 | 0.1 | 0.5 | 5.0 |
| **CVAE** | ✗ | ✓ | ✗ | ✓ | 28.21 | 25.00 | 68.33 | 100.0 |
| | ✓ | ✓ | ✗ | ✓ | 34.98 | 60.00 | 85.00 | 100.0 |
| | ✓ | ✓ | ✓ | ✗ | 38.85 | 5.00 | 30.00 | 100.0 |
| | ✓ | ✓ | ✓ | ✓ | **43.93** | **66.67** | **96.67** | **100.0** |
| **Peptide** | | | | | | | | |
| **Random** | ✗ | ✗ | ✗ | ✓ | 36.55 | 0.1 | 0.5 | 5.0 |
| **CVAE** | ✓ | ✗ | ✗ | ✓ | 53.39 | 3.23 | 5.38 | 25.81 |
| | ✓ | ✓ | ✗ | ✓ | 53.64 | 4.30 | 7.53 | 33.33 |
| | ✓ | ✓ | ✓ | ✗ | 54.02 | 2.15 | 8.60 | 31.18 |
| | ✓ | ✓ | ✓ | ✓ | **61.41** | **11.58** | **22.58** | **67.74** |

**Results**   Table 1 supports the following three key findings: 1) Our contrastive variational autoencoder (CVAE) reliably retrieves relevant interfaces, with RC-5% achieving $100\%$ for antibodies and $67.74\%$ for peptides; 2) Exposure to diverse interface types during training significantly enhances generalization, as incorporating cross-domain training, particularly with intra-protein interfaces from the ProtFrag dataset, improves performance to a large extent; 3) Joint training with both contrastive and VAE losses is essential, as removing the VAE loss (i.e., using an encoder-only model) significantly degrades performance, likely because the reconstruction objective reinforces memorization of meaningful interaction patterns between binding sites and binders. These results demonstrate that our CVAE enables effective retrieval with high recall and precision, forming a solid foundation for retrieval-augmented generation.

## 4.2   Generation Performance

**Metrics**   To evaluate our model's effectiveness in designing peptide–protein and antibody–antigen complexes, we adopt several established structural and biochemical metrics. Amino Acid Recovery (**AAR**) measures sequence fidelity as the proportion of residues in the generated peptide that match the reference, based on canonical sequence alignment [42, 20]. Complex **RMSD** quantifies structural accuracy via the root mean square deviation of $C_\alpha$ coordinates, after aligning binders on the target protein. Interaction Site Match (**ISM**) assesses how well key physicochemical interactions (e.g., hydrogen bonds, hydrophobic contacts) are reproduced, reflecting interface realism [2], computation details are in Appendix A. The change in binding free energy ($\Delta\Delta G$) captures affinity shifts, where negative values indicate stronger binding. **IMP** reports the fraction of targets for which generated binders outperform the native ligand in predicted affinity. For each test complex, we sample 100 candidates per model and evaluate all metrics to ensure statistical robustness.

### 4.2.1   Peptide Sequence-Structure Codesign

**Baselines**   We compare our method against several representative generative frameworks. RFD-iffusion [53] generates peptide backbones, followed by one round of inverse folding using ProteinMPNN [13] and Rosetta-based atomistic relaxation [5]. To avoid overfitting physical metrics,

we restrict this process to a single iteration. We also include peptide-specific models such as PepFlow [36] and PepGLAD [30], which leverage multi-modal flow matching and latent diffusion to jointly model sequence and structure. In addition, we evaluate UniMoMo [31], a unified framework for small molecules, peptides, and antibodies, using its best peptide-generation configuration without incorporating the small molecule dataset.

**Results** As shown in Table 2, our method consistently outperforms all baselines across multiple evaluation metrics. Compared to existing frameworks, our model achieves higher recovery of the ground-truth sequences and structures, effectively leveraging knowledge from existing interfaces. Notably, our **RADiAnce** surpasses baselines by a large margin in terms of ISM, indicating it excels at copying relevant interaction patterns from retrieved interfaces. Further ablation shows improved generation when retrieving cross-domain data over single-domain H.

Table 2: Results for peptide sequence-structure codesign.

| Model | AAR (%) | RMSD (Å) | $\Delta\Delta G$ (kJ/mol) | IMP (%) | ISM (%) |
|---|---|---|---|---|---|
| RFDiffusion | 34.68 | 4.69 | 24.78 | 5.38 | 28.38 |
| PepFlow | 35.47 | 2.87 | 15.71 | 14.13 | 27.83 |
| PepGLAD | 38.62 | 2.74 | 15.26 | 16.13 | 32.63 |
| UniMoMo | 38.69 | 2.31 | 2.409 | 40.86 | 49.13 |
| RADiAnce | **39.42** | **2.29** | **1.963** | **41.94** | **52.15** |

### 4.2.2 Antibody Sequence-Structure Codesign

**Setup** We define Complementarity-Determining Regions (CDRs) according to the Chothia numbering system [12], as adopted in previous literature Luo et al. [40], Kong et al. [31]. For each antigen-antibody complex in the test set, we randomly mask one CDR loop and treat the remaining framework and unmasked CDRs as conditioning context. The model is then tasked with generating the missing CDR regions.

**Baselines** We benchmark our approach against representative state-of-the-art generative models in molecular and antibody design: MEAN [28] uses a multi-channel SE(3)-equivariant GNN to iteratively generate CDR sequences and structures. DyMEAN [29] extends MEAN to full-atom generation and handles uncertain antibody docking poses. DiffAb [40] applies diffusion modeling to jointly generate amino acid types, $C_\alpha$ positions, and orientations. GeoAB [37] combines heterogeneous residue encoders with geometric energy priors for realistic structure generation. UniMoMo [31] unifies the generation of small molecules, peptides, and antibodies within a shared representation framework.

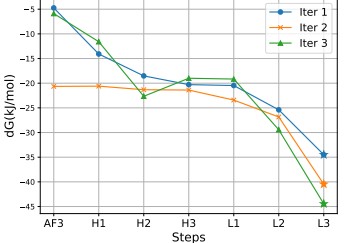

**Results** As shown in Table 3, our method consistently outperforms all baselines on CDR sequence-structure co-design. It achieves notable improvements in sequence recovery, structural accuracy, and binding affinity across all CDR regions, demonstrating the benefits of learning from retrieved interfaces. Moreover, consistent with earlier peptide results, cross-domain retrieval further boosts performance over single-domain retrieval (Appendix H), highlighting the ability of our model to generalize shared interaction patterns across different domains. We also provide a qualitative example in Appendix G, illustrating how the model leverages retrieved interfaces from other domains to correct and enhance generated interactions.

Figure 3: Free energy changes during iterative CDR design.

### 4.3 Antibody Design Without Bound Structural scaffold

To better align with real-world applications, we propose a pipeline for designing all six CDRs of an antibody without access to its bound structure. Beginning with a predefined framework, we iteratively

Table 3: Results for antibody CDRs sequence-structure codesign.

| Model | CDR | AAR | RMSD | $\Delta\Delta G$ | IMP | ISM | CDR | AAR | RMSD | $\Delta\Delta G$ | IMP | ISM |
|---|---|---|---|---|---|---|---|---|---|---|---|---|
| MEAN | | 76.78 | 0.9660 | 3.194 | 20.00 | 66.11 | | 68.84 | 1.075 | 4.828 | 28.30 | 53.42 |
| dyMEAN | | 84.25 | 0.8061 | 18.42 | 1.695 | 56.10 | | 81.28 | 0.8731 | 17.88 | 1.695 | 52.23 |
| GeoAB-R | | 80.00 | 0.5472 | 254.34 | 25.00 | 69.30 | | 79.00 | 0.4670 | 2.757 | 28.33 | 55.13 |
| DiffAb | H1 | 83.25 | 0.6278 | -3.625 | 87.93 | 75.02 | L1 | 83.25 | 0.6278 | -3.424 | 93.10 | 75.02 |
| GeoAB-D | | 85.48 | 0.4290 | _-6.925_ | 83.33 | 74.76 | | 80.65 | 0.3352 | _-6.692_ | 78.33 | 56.32 |
| Unimomo | | _86.72_ | _0.3728_ | -6.247 | _88.33_ | _98.01_ | | 80.28 | _0.3177_ | 2.374 | _93.33_ | _96.37_ |
| RADiAnce | | **90.83** | **0.2977** | **-8.221** | **96.67** | **98.36** | | **86.20** | **0.2907** | **-7.170** | **95.00** | **98.31** |
| MEAN | | 54.16 | 0.9928 | 49.69 | 6.67 | 26.98 | | 64.24 | 0.7657 | 13.08 | 10.00 | 72.37 |
| dyMEAN | | 73.62 | 0.9453 | 18.45 | 3.390 | 48.35 | | 84.35 | 0.7514 | 17.58 | 0.00 | 59.41 |
| GeoAB-R | | 54.21 | 0.7125 | 30.23 | 16.67 | 27.67 | | 75.67 | 0.4742 | 5.691 | 35.00 | 72.58 |
| DiffAb | H2 | 67.51 | 0.4378 | -1.895 | 77.58 | 29.11 | L2 | 72.87 | 0.3910 | **-8.397** | 86.21 | 72.12 |
| GeoAB-D | | 75.04 | 0.4828 | 3.982 | 56.67 | 28.95 | | 83.06 | 0.3462 | -5.067 | 76.67 | 73.16 |
| Unimomo | | _75.74_ | _0.2207_ | **-2.710** | **88.33** | _86.31_ | | _84.36_ | _0.1592_ | _-4.779_ | 95.00 | _96.55_ |
| RADiAnce | | **79.20** | **0.2135** | _-2.370_ | 83.33 | **87.83** | | **86.68** | **0.1558** | -3.857 | 95.00 | **98.52** |
| MEAN | | 26.19 | 1.076 | 104.88 | 8.30 | 12.81 | | 53.72 | 1.100 | 9.835 | 18.33 | 49.52 |
| dyMEAN | | 31.65 | 1.123 | 24.23 | 1.695 | 41.67 | | _72.86_ | 0.8722 | 18.39 | 3.390 | 57.16 |
| GeoAB-R | | 32.04 | 1.682 | 42.71 | 0.33 | 11.67 | | 64.16 | 0.7213 | 5.161 | 35.00 | 51.27 |
| DiffAb | H3 | 51.84 | 1.584 | -1.580 | 56.90 | 27.40 | L3 | 64.16 | 0.8005 | -6.685 | 86.21 | 57.70 |
| GeoAB-D | | 44.82 | 1.565 | 34.73 | 41.67 | 22.50 | | 64.16 | 0.8005 | -3.445 | 80.00 | 54.76 |
| Unimomo | | _52.34_ | _1.040_ | **-7.438** | _65.00_ | _71.52_ | | 72.36 | _0.5003_ | _-8.797_ | **95.00** | _90.94_ |
| RADiAnce | | **54.66** | **0.9443** | _-6.236_ | **71.67** | **71.64** | | **76.75** | **0.4478** | **-9.207** | _90.00_ | **94.82** |

alternate between CDR redesign using our **RADiAnce** and structure prediction with AlphaFold3 [1], progressively refining both the sequence and structure toward an optimal binder.

To design a novel antibody targeting the HIV-1 primary receptor CD4 (PDBID: 3O2D) [15], we use Ramucirumab (DrugBank ID: DB05578) as the structural scaffold, which originally targets VEGFR2 [24] but with good drug developability. We identify the CD4 epitope via AlphaFold3-guided docking [1], and iteratively redesign the complementarity-determining regions (CDRs) by knocking out and regenerating each loop using our model. After each iteration, the antibody is relaxed with Rosetta [5] and re-docked against CD4. This redesign–dock cycle is repeated three times to progressively refine binding. We use PyRosetta to estimate binding free energy ($\Delta G$), considering

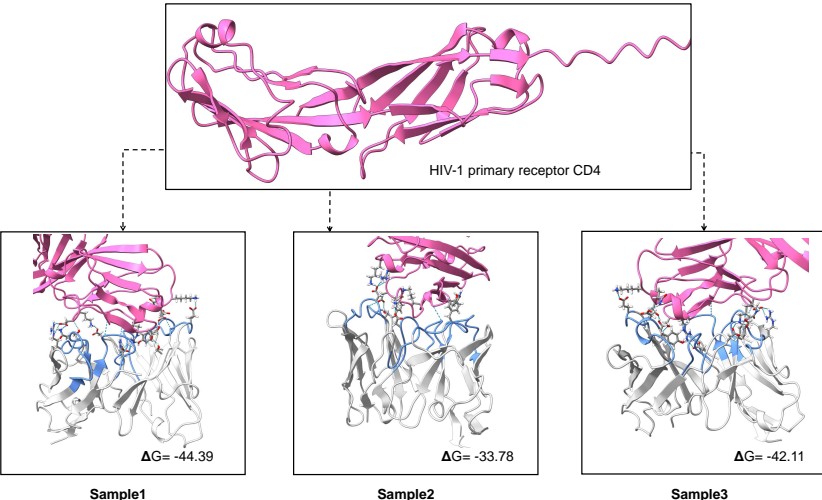

Figure 4: Examples of de novo antibody designs targeting the HIV-1 receptor CD4. Each case shows the final docked complex with the redesigned antibody interacting with the target epitope.

designs with $\Delta G < 0$ as indicative of favorable antigen engagement, even without a fixed antibody framework. We visualize the final antibody–antigen complex and highlight hydrogen bonds at the interface (Figure 4). Figure 3 illustrates the $\Delta G$ trajectory over design iterations, showing consistent affinity improvements, with the final design reaching $\Delta G < 0$. These results highlight the potential

Table 4: Comparison of retrieval strategies for HCDR3 and peptide codesign tasks.

| Strategy | Retrieved | HCDR3 | | | | Peptide | | | |
|---|---|---|---|---|---|---|---|---|---|
| | | Similarity | AAR | RMSD | Diversity | Similarity | AAR | RMSD | Diversity |
| Top-N | 0 | N/A | 51.55 | 0.9525 | 0.0524 | N/A | 37.94 | 2.38 | 0.5042 |
| Top-N | 1 | 57.62±5.3 | 53.26 | **0.9431** | 0.0404 | 55.56±4.5 | 39.03 | 2.38 | 0.4846 |
| Top-N | 10 | 54.65±4.9 | **54.66** | 0.9443 | 0.0484 | 51.67±3.6 | 39.12 | 2.31 | 0.5129 |
| Top-N | 20 | 50.43±4.8 | 54.63 | 0.9682 | 0.0485 | 50.24±3.3 | **39.42** | 2.29 | 0.5227 |
| Top-N | 40 | 46.04±4.7 | 54.06 | 0.9653 | 0.0494 | 47.68±3.1 | 38.16 | **2.28** | 0.5273 |
| Reverse-N | 10 | -49.86±16.7 | 52.31 | 0.9718 | 0.0468 | -26.09±12.5 | 36.33 | 3.09 | 0.5267 |
| Random | 10 | 17.34±6.9 | 51.72 | 0.9665 | **0.0649** | 11.78±8.16 | 38.04 | 2.56 | **0.5580** |

Table 5: Performance with adaptive retrieval for HCDR3 and Peptide codesign tasks.

| Settings | AAR (%) | RMSD (Å) | Diversity | $\Delta\Delta G$ (kJ/mol) | FoldX $\Delta G$ (kJ/mol) | ISM (%) | DockQ |
|---|---|---|---|---|---|---|---|
| **HCDR3** | | | | | | | |
| Fixed Top-N | 54.63 | 0.9682 | 0.0485 | -6.236 | -8.4250 | 71.64 | 0.9582 |
| Adaptive Retrieval | **54.71** | **0.9147** | **0.0523** | **-8.938** | **-8.6180** | 71.61 | **0.9604** |
| **Peptide** | | | | | | | |
| Fixed Top-N | 39.42 | 2.29 | 0.5227 | 1.963 | -7.99 | 52.15 | 0.7698 |
| Adaptive Retrieval | 39.22 | **2.27** | **0.5400** | **1.427** | -7.97 | **52.89** | **0.7762** |

of our framework to generate *de novo* antibodies capable of strong and specific epitope binding without requiring a pre-solved complex.

## 5 Analysis

### 5.1 Effect of Retrieval Quantity and Quality

To better understand how retrieval influences our model performance, we evaluate **RADiAnce** under different retrieval modes and varying numbers of retrieved samples on codesign tasks.

**Results** Table 4 summarizes the relationship between retrieval strategy and model performance. Increasing the number of retrieved samples generally improves performance, but excessively large pools introduce low-similarity samples that add noise and degrade quality. Random and reverse retrieval perform poorly for the same reason, while the no-retrieval baseline is markedly worse than any retrieval-based setting, highlighting the importance of retrieval. We also find that larger pools and the no-retrieval case yield higher diversity, whereas retrieving only one sample produces the lowest. This aligns with our expectation, as limited retrieval can overly constrain the generative space.

### 5.2 Adaptive Retrieval Approach

Building on the results above, we replace fixed top-$N$ retrieval with an adaptive similarity cutoff: only neighbors above a task-specific threshold are retained, allowing the model to automatically use more samples when many are relevant and fewer when they are not. As shown in Table 5, this adaptive strategy consistently improves multiple metrics by preserving the benefits of larger retrieval pools while avoiding noise from less relevant neighbors.

## 6 Conclusions and Limitations

We propose **RADiAnce**, a retrieval-augmented framework that unifies retrieval and generation in a contrastive latent space to improve rational and interpretable binder design. A contrastive VAE enables efficient cross-domain motif retrieval, while cross-attention-guided latent diffusion enhances sequence–structure codesign. Despite its effectiveness, performance depends on retrieval quality, motivating future work on improving structural descriptors and exploring more robust and structure-aware conditional integration strategies. Beyond methodological advances, our framework enables a transparent and interpretable binder design process, potentially accelerating biologic development through more controllable generation of therapeutically meaningful molecules.

## Acknowlegements

This work is jointly supported by the National Key R&D Program of China (No.2022ZD0160502), the National Natural Science Foundation of China (No. 61925601, No. 62376276, No. 62276152), and Beijing Nova Program (20230484278).

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

## A    Implementation of Interaction Site Match (ISM)

The **Interaction Site Match (ISM)** metric evaluates the fine-grained accuracy of predicted molecular interactions by assessing their strict consistency with the ground truth at the atomic level [2]. An interaction is considered a strict match only when the predicted interaction has the same interaction type (such as hydrogen bond or salt bridge), originates from the same residue on the target and terminates at the same residue as in the reference. To compute ISM, we iterate over all annotated residue-level interactions in the reference structure and search for corresponding interactions in the predicted structure. If a predicted interaction matches a reference one in both ligand atom and interaction type, it is counted as a match. The final ISM score is calculated as the ratio between the number of strictly matched interactions and the total number of interactions in the reference:

In cases where the reference structure contains no annotated interactions (i.e., the denominator is zero), the ISM score is defined as `NaN` and excluded from downstream averaging. This prevents distortion of the overall metric by non-informative cases and ensures that the evaluation focuses on structurally meaningful interaction sites. ISM thus captures the model's ability to recover detailed and chemically precise binding patterns at the atomic level.

## B    Implementation of Interaction Type Overlap (ITO)

The **Interaction Type Overlap (ITO)** metric measures the global agreement in interaction type distributions between the predicted and reference complexes, regardless of specific atom-level correspondences [2]. It focuses on comparing the overall frequency of each interaction type, rather than their precise locations.

To compute ITO, we count the number of occurrences of each interaction type in both the reference and predicted structures. For each interaction type, the overlap is defined as the minimum of the two counts. The ITO score is then calculated as the total overlap across all interaction types, normalized by the total number of interactions in the reference:

$$\text{ITO} = \frac{\sum_{\text{type}} \min(\text{count}_{\text{ref}}, \text{count}_{\text{pred}})}{\sum_{\text{type}} \text{count}_{\text{ref}}}$$

Similarly, if the reference structure contains no interactions, the ITO score is set to `NaN` and omitted from any aggregate analysis. This design ensures that only informative samples contribute to dataset-level evaluations. ITO captures the model's ability to reproduce the overall molecular interaction profile and is especially useful for assessing chemically meaningful binding preferences at a global scale.

## C    Definition of Diversity Metrics

**Diversity** quantifies the diversity of generated peptides as the ratio of unique clusters to total generations. Sequence and structure clustering thresholds are set at sequence identity above 40% and RMSD below 2 Å, respectively. For instance, a sequence diversity of 0.0593 for HCDR3 means that 100 generated sequences form approximately 6 distinct clusters. The diversity metrics for our model are presented in Table 6.

Table 6: Diversity of generated samples

| Settings | Sequence Diversity |
| --- | --- |
| MEAN (HCDR3) | 0.0100 |
| RADiAnce (HCDR3) | **0.0593** |
| Pepflow | 0.0745 |
| RADiAnce (Peptide) | **0.558** |

The results demonstrate that diffusion based methods achieve a higher degree of sequence diversity compared to non diffusion based methods.

# D Ablations on Condition Integration Strategy

## D.1 AdaLN-Zero Conditioning Mechanism

We implement the conditional noise prediction network $\epsilon_\theta(\mathcal{Z}_x^t, \mathcal{Z}_y, \mathbb{T}^v, t)$ by extending the Equivariant Full-Atom Transformer architecture [25] with AdaLN-Zero conditioning [44]. Each layer simultaneously updates the E(3)-invariant scalar latent features $\mathcal{Z}_t$ and vector latent features $\boldsymbol{v}_t$ through attention and feed-forward stages, all equipped with adaptive layer normalization and residual scaling.

Let $\boldsymbol{v}_i \in \mathbb{R}^{3 \times m \times h}$ denote the vector features corresponding to $\mathcal{Z}_i$, and let $\boldsymbol{T} \in \mathbb{R}^{n \times h}$ be the conditioning input. The update rules at layer $l$ are:

$$\mathcal{Z}_t^{(l-0.5)}, \boldsymbol{v}_t^{(l)} = \mathcal{Z}_t^{(l-1)} + \alpha_{\text{attn}} \cdot \text{SelfAttn}(\text{AdaLN}(\mathcal{Z}_t^{(l-1)}, \boldsymbol{T}), \boldsymbol{v}_t^{(l-1)}), \tag{13}$$

$$\mathcal{Z}_t^{(l)} = \mathcal{Z}_t^{(l-0.5)} + \alpha_{\text{ffn}} \cdot \text{FFN}(\text{AdaLN}(\mathcal{Z}_t^{(l-0.5)}, \boldsymbol{T}), \boldsymbol{v}_t^{(l-1)}), \tag{14}$$

where $\text{AdaLN}(\cdot, \boldsymbol{T})$ applies zero-initialized, conditioning-aware layer normalization to the input, and $\alpha_{\text{attn}}, \alpha_{\text{ffn}} \in \mathbb{R}^{1 \times h}$ are scaling factors computed as:

$$\boldsymbol{\alpha} = f_{\text{scale}}\left(\frac{1}{n}\sum_{j=1}^{n}\boldsymbol{T}^{(j)}\right). \tag{15}$$

Here, $f_{\text{scale}} : \mathbb{R}^h \to \mathbb{R}^h$ is a learnable linear transformation initialized to zero, responsible for producing the residual scaling weights.

This conditioning scheme allows each layer to modulate feature updates based on template $\boldsymbol{T}$, while ensuring stable training via identity initialization and structured residuals.

## D.2 Contextual Prompt Integration

To further enhance conditional control, we implement a context-aware integration layer within each transformer block. This design leverages prompt features $\boldsymbol{T}$ (e.g., interface representations retrieved from a database) to modulate the latent trajectory. The integration process involves attending to the prompt features to select the most relevant context and then fusing it with the latent representation. The update of scalar and vector features follows three stages:

Let $\mathcal{Z}_t^{(l-1)} \in \mathbb{R}^{m \times h}$ and $\boldsymbol{T} \in \mathbb{R}^{n \times h}$ denote the input features and prompt templates, respectively. At transformer layer $l$, the update rules are:

$$\boldsymbol{T}^{\text{sel}} = \text{Softmax}\left(\frac{\boldsymbol{Q}\boldsymbol{K}^\top}{\sqrt{h}}\right)\boldsymbol{T}, \quad \boldsymbol{Q} = \mathcal{Z}_t^{(l-1)}\boldsymbol{W_Q}, \quad \boldsymbol{K} = \boldsymbol{T}\boldsymbol{W_K}, \tag{16}$$

$$\mathcal{Z}_t^{(l-0.5)} = \text{Fuse}\left(\mathcal{Z}_t^{(l-1)}, \boldsymbol{T}^{\text{sel}}\right), \tag{17}$$

$$\left[\mathcal{Z}_t^{(l)}, \boldsymbol{v}_t^{(l)}\right] = \text{FFN}\left(\text{SelfAttn}\left(\mathcal{Z}_t^{(l-0.5)}, \boldsymbol{v}_t^{(l-1)}\right)\right), \tag{18}$$

where $\boldsymbol{W_Q}, \boldsymbol{W_K} \in \mathbb{R}^{h \times h}$ are learnable projections. The attention mechanism computes similarity scores between latent tokens and prompt features to produce a soft selection $\boldsymbol{T}^{\text{sel}}$, enabling each latent position to selectively attend to the most informative prompts. The fusion function $\text{Fuse}(\cdot, \cdot)$ denotes a multilayer perceptron operating on the concatenation $[\mathcal{Z}_t^{(l-1)}\|\boldsymbol{T}^{\text{sel}}]$, projecting it back to the original dimension $h$.

This mechanism enables fine-grained control over the generation process by allowing each latent token to dynamically attend to prompt embeddings and modulate its representation accordingly.

Table 7: Results for antibody CDR-H3 sequence-structure codesign.

| Fusion method | AAR (%) | RMSD (Å) | $\Delta\Delta G$(kJ/mol) | IMP (%) | ISM (%) |
|---|---|---|---|---|---|
| AdaLN-Zero | 50.42 | 0.964 | -7.047 | 68.33 | **73.69** |
| In-context | 51.03 | 1.019 | **-8.102** | 66.67 | 72.57 |
| Cross Attention | **54.66** | **0.9443** | -6.236 | **71.67** | 71.64 |

## D.3 Results

As shown in Table 7, the Cross Attention fusion mechanism demonstrates the most effective performance among all compared methods. It consistently achieves the best overall results across sequence accuracy, structural alignment, and interaction quality. This suggests that cross-attention enables more precise integration of contextual and structural information, leading to improved sequence-structure codesign for antibody CDRs. Its superior performance highlights the importance of flexible and fine-grained feature fusion in guiding generative antibody modeling.

# E Implementation Details

## E.1 Baselines

**Peptide** For **RFdiffusion**, since the training pipeline for custom datasets is not publicly available, we directly use the official pretrained weights and inference scripts for binder design. For **PepGLAD** and **PepFlow**, we adopt the official implementations and retrain them on our peptide dataset, using the default hyperparameters provided in their repositories. For **UniMoMo**, we retrain the model on the same cross-domain dataset as our method, using the hyperparameter settings provided in the original publication [31]. All models are evaluated on the same dataset to ensure fair comparison.

**Antibody** For **MEAN**, **DyMEAN**, **DiffAb**, **GeoAB-R**, and **GeoAB-D**, we use their official implementations and retrain the models on the same antibody dataset as our model, using the default hyperparameters specified in their repositories. For **UniMoMo**, we retrain the model on the same cross-domain dataset as our method, using the hyperparameter settings provided in the original publication [31]. To ensure consistency, we convert all antibody sequences from IMGT [33] to Chothia numbering [12], following the protocol of **DiffAb** [40]. The Chothia system provides stricter and more structure-consistent definitions of complementarity-determining regions (CDRs). This conversion may lead to a drop in amino acid recovery (AAR) compared to IMGT, as the Chothia system avoids overestimating recovery due to trivial unigram patterns [31]. For **GeoAB**, since the official implementation only supports HCDR3 prediction, we extend it to cover additional CDRs following the same processing strategy. Training samples that fail preprocessing are excluded. All models are evaluated on the same dataset to ensure fair comparison.

## E.2 RADiAnce

We trained our model using 8 GPUs with 80 GB memory in parallel. The hyperparameter configurations for both Contrastive VAE and Diffusion models are summarized in Table 8.

# F Comparison Between Generated Samples and Retrieved Exemplars

To quantitatively assess how well the generated interfaces recapitulate the interaction patterns of their retrieval-based conditioning, we report two key metrics: Interaction Type Overlap with the Reference (ITO-Gen) and Interaction Type Overlap with the Retrieved Exemplar (ITO-RAG), evaluated for both antibody HCDR3 and peptide cases (Figure 5).

Specifically, ITO-Gen measures the degree of overlap in interaction patterns between the generated samples and their corresponding reference complexes, reflecting the fidelity of binding mode reconstruction. In contrast, ITO-RAG quantifies the similarity between the generated samples and the retrieved exemplars, indicating the effectiveness of retrieval in guiding the generative process.

Table 8: Hyperparameters of **RADiAnce**

| Name | Configuration | Description |
|---|---|---|
| *Contrastive VAE* | | |
| Encoder / Decoder Type | EPT | Backbone architecture for encoder/decoder |
| latent_size | 8 | Dimension of latent state |
| hidden_size | 512 | Feature dimensionality for node and edge embeddings |
| edge_size | 64 | Size of edge type embeddings |
| n_layers | 6 | Transformer depth in encoder/decoder |
| n_heads | 8 | Number of heads for multihead self attention per layer |
| k_neighbors | 9 | Graph connectivity for spatial features |
| cutoff | 10.0Å | Distance threshold for RBF kernels |
| KL_loss_weights | 0.6 / 0.8 | Weight for KL divergence of structure / sequence latents |
| atom_coord_loss_weights | 1.0 | Weight for atom coordinate loss |
| block_type_loss_weights | 1.0 | Weight for categorical loss on block (residue) types |
| contrastive_loss_weights | 1.0 | Weight for contrastive similarity loss |
| local_distance_loss_weights | 0.5 | Weight for intra-structure distance loss |
| bond_loss_weights | 0.5 | Weight for bond type classification loss |
| *Conditional Latent Diffusion* | | |
| hidden_size | 512 | Dimension of hidden states |
| T | 100 | Diffusion steps |
| n_layers | 6 | Number of denoising layers |
| n_heads | 8 | Number of heads for multihead self and cross attention |
| n_rbf | 64 | Number of RBF kernels |
| cutoff | 3.0Å | Cutoff distance for RBF kernels |

The overlap with reference complexes reflects the model's ability to reconstruct native binding modes, while the overlap with retrieved exemplars assesses how effectively information from retrieval guides generation. Our results indicate that the generated molecular interfaces capture a substantial proportion of the chemically meaningful interactions present in the reference structures—achieving, for example, 60.7% overlap for HCDR3 and 65.0% for peptide cases. At the same time, the retrieved exemplars themselves provide strong and relevant interaction patterns that align closely with the reference, offering reliable and informative guidance for the generative process.

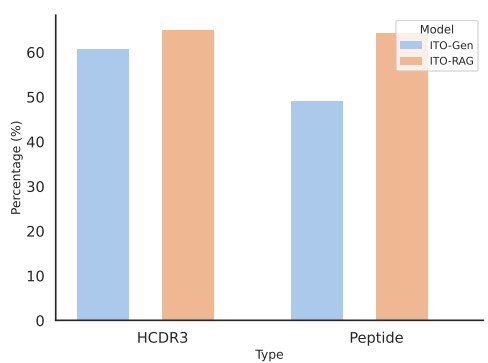

Figure 5: Interaction overlap analysis.

These results collectively demonstrate that our retrieval-augmented framework, by leveraging structurally relevant examples, enables the effective transfer and reconstruction of key interaction motifs, thus enhancing the chemical rationality and interpretability of the designed interfaces.

## G Case Study: Qualitative Improvement Through Retrieval Conditioning

GPIIb/IIIa (PDB ID: 3NID) is a key platelet integrin that mediates adhesion and aggregation during thrombus formation [56]. In the 3NID structure, it adopts a closed headpiece conformation when bound to a specific antagonist that stabilizes the inactive state and prevents activation. As illustrated in Figure 6, we investigate the design of the HCDR3 loop for the GPIIb/IIIa binder under two settings: unconditional generation and retrieval-augmented generation. Without retrieval guidance, the model struggles to reconstruct the characteristic multi-hydrogen bond interactions observed in the reference structure. In contrast, retrieval from the database yields top-10 candidates that include two peptide complexes and one antibody-antigen complex, all of which exhibit similar interaction patterns to the target. Specifically, these retrieved structures share the distinctive formation of hydrogen bonds

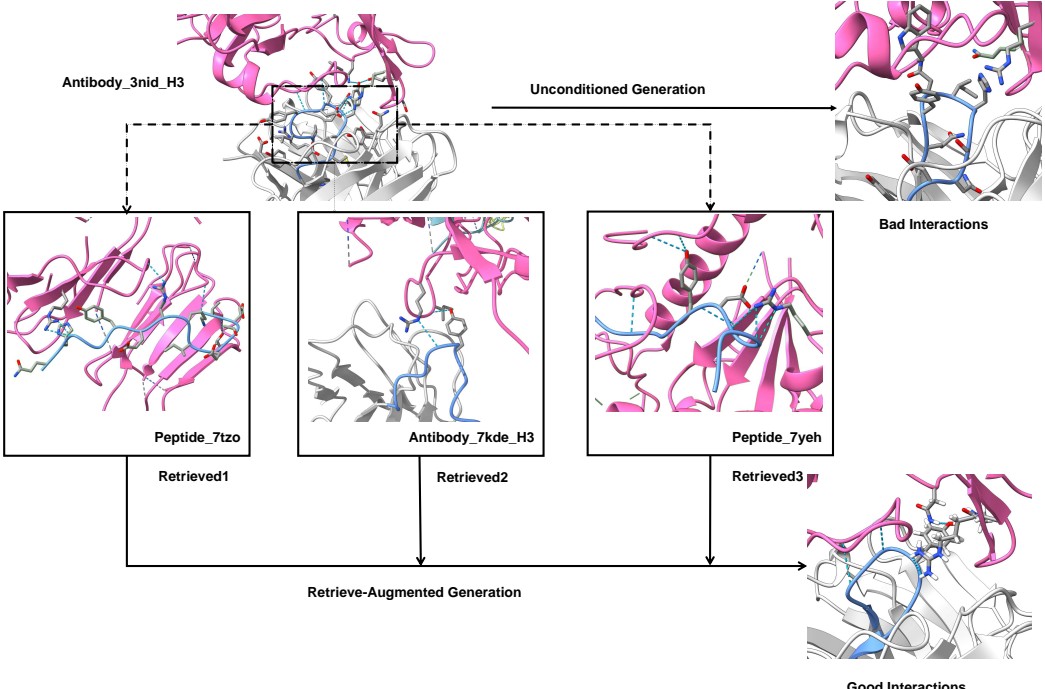

Figure 6: Case study of HCDR3 design for the GPIIb/IIIa(PDBID:3NID) binder. The binding site is shown in pink, binder key residues in blue, and hydrogen bonds in light blue dashed lines. Guided by retrieved exemplars, retrieval-augmented generation successfully preserves these key interaction modes.

between an arginine residue at the binding site and the binder, as well as between a tyrosine residue in the binder and the binding site. Consequently, the RAG-based generation inherits these critical interaction motifs, successfully recovering the hydrogen-bonding patterns mediated by arginine and tyrosine residues. This case exemplifies how our retrieval-augmented generation framework effectively leverages retrieved interaction patterns to guide and enhance molecular design.

## H Cross-Domain Retrieval Ablation Studies

To further elucidate the impact of cross-domain retrieval on our model's generative performance, we conducted a series of ablation studies by selectively restricting the set of available templates during inference. Specifically, we compared the outcomes of conditioning the generation process using retrieval exemplars sourced from either a single domain (e.g., antibody-only, peptide-only, etc) versus those retrieved from the entire, cross-domain template pool.

The results in Table 9 and Table 10 reveal a consistent trend: models leveraging cross-domain retrieval demonstrate superior capability in reconstructing chemically and geometrically plausible interface interactions, as measured by both interaction recovery metrics and binding affinity scores. In contrast, restricting the retrieval to templates within the same domain as the query generally leads to diminished performance, manifesting as reduced interaction overlap and less robust generalization to diverse binding interfaces.

These results highlight the critical advantage of our approach: by comprehensively referencing interface templates across distinct molecular domains, our model can flexibly exploit shared principles of molecular recognition and binding site architecture, thereby enabling more rational and reliable binder generation. The empirical gains observed across all tested scenarios underscore the necessity of cross-domain template integration as a core design choice for retrieval-augmented generative frameworks.

Table 9: Results of recovery metrics for RADiAnce under different template sources on antibody cdrs codesign task. Columns "Ab", "Pep", and "ProtFrag" indicate whether the respective template domain is included for retrieval during inference (✓ for included, ✗ for excluded).

| CDR | Ab | Pep | ProtFrag | AAR (%) | RMSD (Å) | $\Delta\Delta G$ (kJ/mol) | IMP (%) | ISM (%) |
|---|---|---|---|---|---|---|---|---|
| | ✓ | ✗ | ✗ | 90.54 | **0.295** | -7.955 | **98.33** | **98.36** |
| H1 | ✓ | ✓ | ✗ | 90.35 | 0.3007 | -7.705 | 95.00 | **98.36** |
| | ✓ | ✓ | ✓ | **90.83** | 0.2977 | **-8.221** | 96.67 | **98.36** |
| | ✓ | ✗ | ✗ | 78.64 | 0.2157 | -2.605 | **88.33** | 87.83 |
| H2 | ✓ | ✓ | ✗ | 78.70 | 0.2158 | **-2.635** | 80.0 | **88.10** |
| | ✓ | ✓ | ✓ | **79.20** | **0.2135** | -2.370 | 83.33 | 87.83 |
| | ✓ | ✗ | ✗ | 54.47 | 0.9475 | -5.200 | 65.0 | 71.96 |
| H3 | ✓ | ✓ | ✗ | 54.16 | 0.9557 | -4.545 | 66.7 | **72.81** |
| | ✓ | ✓ | ✓ | **54.66** | **0.9443** | **-6.236** | **71.67** | 71.64 |
| | ✓ | ✗ | ✗ | **86.48** | 0.2902 | **-7.59** | **96.67** | 97.88 |
| L1 | ✓ | ✓ | ✗ | **86.48** | **0.2887** | -7.50 | 95.00 | 97.64 |
| | ✓ | ✓ | ✓ | 86.20 | 0.2907 | -7.170 | 95.00 | **98.31** |
| | ✓ | ✗ | ✗ | **87.13** | **0.1550** | -4.435 | **96.67** | 98.52 |
| L2 | ✓ | ✓ | ✗ | 86.99 | 0.1553 | **-4.647** | 95.00 | 98.52 |
| | ✓ | ✓ | ✓ | 86.68 | 0.1558 | -3.857 | 95.00 | 98.52 |
| | ✓ | ✗ | ✗ | 76.71 | 0.4398 | -5.94 | 86.67 | **95.08** |
| L3 | ✓ | ✓ | ✗ | 76.41 | **0.4372** | -7.43 | **93.33** | **95.08** |
| | ✓ | ✓ | ✓ | **76.75** | 0.4478 | **-9.207** | 90.00 | 94.82 |

Table 10: Results of recovery metrics for RADiAnce under different template sources on peptide codesign task. Columns "Ab", "Pep", and "ProtFrag" indicate whether the respective template domain is included for retrieval during inference (✓ for included, ✗ for excluded).

| Ab | Pep | ProtFrag | AAR (%) | RMSD (Å) | $\Delta\Delta G$(kJ/mol) | IMP (%) | ISM (%) |
|---|---|---|---|---|---|---|---|
| ✗ | ✓ | ✗ | 38.97 | **2.27** | 3.424 | 40.86 | 49.82 |
| ✓ | ✓ | ✗ | **39.60** | 2.41 | 2.423 | 38.71 | 48.33 |
| ✓ | ✓ | ✓ | 39.42 | 2.29 | **1.963** | **41.94** | **52.15** |

# I  Ablation on Single-Domain Data

To ensure that the performance comparison is not biased by the use of multiple binder types, we conducted an ablation study where both the training and retrieval data were restricted to a single domain. Specifically, we evaluated **RADiAnce** and UniMoMo under the antibody heavy chain CDR3 (AbH3) and peptide design tasks, each trained and retrieved using data from only that domain. This setup removes cross-type information and ensures a strictly matched comparison between the models.

Table 11: Ablation study on single-domain data. Both training and retrieval were performed within the same domain to ensure a fair comparison. Other single-domain results are omitted as they are the same result with those reported in the main text.

| Task and models | AAR (%) | RMSD (Å) | $\Delta\Delta G$ (kJ/mol) | IMP (%) | ISM (%) |
|---|---|---|---|---|---|
| UniMoMo (AbH3) | 48.78 | 1.39 | -5.781 | 63.33 | 65.46 |
| RADiAnce (AbH3) | **51.31** | **1.109** | **-5.994** | **68.33** | **69.71** |
| UniMoMo (Peptide) | 37.59 | 2.48 | 7.69 | 29.03 | 40.08 |
| RADiAnce (Peptide) | **38.28** | **2.37** | **7.57** | 27.95 | **45.37** |

As shown in Table 11, even when trained and retrieved within a single domain, **RADiAnce** consistently outperforms other models across evaluated metrics. This demonstrates that the performance gain of **RADiAnce** is not solely due to multi-type data retrieval, but rather arises from its intrinsic ability to model contextually aligned binder–target interactions.

## J  Discussion

### J.1  Reproducibility and Statistical Robustness

To demonstrate the robustness and reliability of our evaluation, we explicitly quantify the statistical variability of our model performance under different random initializations. Our main results are accompanied by standard deviations that measure model variability across independent runs with different random seeds. Specifically, we reran the entire inference and evaluation pipeline three times with distinct random seeds, and computed the mean and standard deviation across these runs. Given that each evaluation aggregates thousands of test instances, the per-run variation remains small, indicating robustness and reproducibility of our evaluation benchmark.

Table 12: Performance metrics with standard deviations across independent runs. The consistently small deviations confirm the stability of our evaluation.

| Model | AAR (%) | RMSD (Å) | $\Delta\Delta G$ (kJ/mol) | IMP (%) | ISM (%) |
|-------|---------|----------|---------------------------|---------|---------|
| HCDR3 | 54.66±0.0026 | 0.9443±0.0156 | $-6.236 \pm 0.5862$ | 71.67±0.0096 | 71.64±0.0085 |
| Peptide | 39.42±0.0014 | 2.29±0.0043 | $1.963 \pm 0.2302$ | 41.94±0.0108 | 52.15±0.0007 |

As shown in Table 12, the standard deviations across independent runs are consistently small across all metrics. This confirms that the evaluation is highly stable, with negligible sensitivity to random initialization or stochastic effects during inference.

### J.2  Extended Evaluation Metrics

To provide a more comprehensive evaluation of our model, we further introduced several robust and widely used metrics, including DockQ, FoldX $\Delta G$, and Binding Site Recovery. These metrics complement the previously reported ones by assessing structural and energetic aspects of the predicted complexes.

Table 13: Extended evaluation metrics for RADiAnce and UniMoMo on antibody and peptide datasets.

| Model | DockQ | FoldX $\Delta G$ (kJ/mol) | Binding Site Recovery |
|-------|-------|---------------------------|------------------------|
| RADiAnce (AbH3) | 0.9582 | -8.4250 | 0.9964 |
| UniMoMo (AbH3) | 0.9491 | -7.9950 | 0.9962 |
| RADiAnce (Peptide) | 0.7698 | -7.9891 | 0.9857 |
| UniMoMo (Peptide) | 0.7592 | -7.4901 | 0.9822 |

**DockQ**  DockQ is a continuous quality score for protein–protein docking models ranging from 0 to 1. It integrates the fraction of native contacts ($F_{nat}$), ligand RMSD (LRMSD), and interface RMSD (iRMSD) to provide an overall measure of structural agreement with the native complex. Scores above 0.23, 0.49, and 0.80 correspond to acceptable, medium, and high-quality models, respectively [7].

**FoldX $\Delta G$**  FoldX calculates the binding free energy ($\Delta G$) of a protein–protein complex as the difference between the Gibbs free energy of the complex and the sum of its unbound partners. More negative $\Delta G$ values indicate stronger and more stable interactions [9].

**Binding Site Recovery**  Binding Site Recovery measures the proportion of true interface residues correctly identified by the model, representing the accuracy of interface residue prediction. Higher values indicate that the generated complex preserves the biologically relevant interaction sites.

As shown in Table 13, the newly introduced metrics are consistent with the previously reported performance trends. **RADiAnce** outperforms UniMoMo across both antibody and peptide benchmarks, achieving higher DockQ scores and more favorable FoldX $\Delta G$. These results confirm the robustness and reliability of our model across diverse evaluation metrics.

### J.3 Dataset Analysis and Benchmark Reliability

Data leakage in retrieval-augmented generative frameworks can lead to overly optimistic results if test samples share high similarity with retrieved templates. To address this concern, we performed a comprehensive verification of our data split.

**Established conventions and potential issues**   Following prior studies in antibody and peptide design, we adopt the RABD and LNR benchmarks, respectively, and enforce a 40% sequence-identity clustering to prevent data leakage across train, validation, and test partitions. However, as pointed out in prior discussions, sequence clustering alone may be insufficient, since structural similarity (e.g., TM-score) can still introduce hidden overlaps across sets.

**Verification of data split integrity**   To rigorously verify the split, we computed both sequence identity and TM-score between each test sample and the entire training set (Table 14). The overall low similarity confirms that the test data are largely distinct from the training examples. Nonetheless, a few rare cases with **TM-score > 0.50** were observed, indicating that minor structural overlaps may still occur under conventional data-splitting method.

Table 14: Verification of data-split integrity. All values represent mean $\pm$ standard deviation.

| Test Set | TM-score with Train Set | SeqId with Train Set |
|----------|------------------------|----------------------|
| RABD     | $0.2421 \pm 0.053$     | $0.1272 \pm 0.021$   |
| LNR      | $0.2503 \pm 0.069$     | $0.0983 \pm 0.017$   |

**Re-evaluation after removing overlapping cases**   To completely remove any potential confounding factor, we excluded any test sample that had at least one training neighbor with a TM-score > 0.50. On the LNR dataset, this excluded 43 samples, leaving 50 for re-evaluation. Table 15 summarizes the resulting performance across multiple baselines.All models experienced performance degradation, which may result from both inadvertent data overlap in previous works and the inherently more challenging or biased characteristics of the remaining test cases. Importantly, our model (**RADiAnce**) not only maintained the strongest overall performance after this correction but also exhibited the smallest performance drop among all methods, further demonstrating that its advantage does not stem from data leakage but from genuinely robust retrieval guided generation.

Table 15: Baseline performance drop after removing overlapping test cases (TM-score > 0.50). Numbers in parentheses show relative change; lower RMSD/$\Delta\Delta G$ and higher AAR/ISM are better.

| Model | AAR (%) | RMSD (Å) | $\Delta\Delta G$ (kJ/mol) | ISM (%) |
|-------|---------|----------|---------------------------|---------|
| PepFlow  | 31.41 (-11.4%) | 3.63 (+26.6%) | 18.84 (+19.9%) | 22.22 (-21.7%) |
| PepGLAD  | 33.85 (-12.4%) | 3.43 (+25.2%) | 18.66 (+22.3%) | 27.67 (-15.2%) |
| UniMoMo  | 35.00 (-9.36%) | 2.87 (+24.2%) | 3.52 (+46.1%) | 38.69 (-21.2%) |
| RADiAnce | **35.79** (-9.20%) | **2.80** (+22.3%) | **2.21** (+12.7%) | **42.26** (-19.0%) |

**Implications for future benchmarks**   This analysis reveals that even widely used benchmarks may inadvertently allow structural redundancy across data splits. Our findings suggest that future dataset construction should incorporate stricter filtering, combining both sequence and structure based similarity constraints (e.g., TM-score thresholds) to ensure fair evaluation. We encourage the community to establish standardized, publicly verifiable data splitting protocols to enhance reproducibility and avoid unintended information leakage in further research.

## K   Code Availability

The source code of the **RADiAnce** is available at `https://github.com/srhn225/RADiAnce`.

