# OpenReview forum: "Latent Retrieval Augmented Generation of Cross-Domain Protein Binders"
_NeurIPS.cc/2025/Conference — NeurIPS 2025 poster_

### Official Review · Reviewer_KBot · 2025-06-14

**Clarity:** 3
**Significance:** 2
**Originality:** 2
**Rating:** 4
**Confidence:** 4

**Summary:**

The paper introduces **Retrieval-Augmented Diffusion for Aligned Interfaces (RADiAnce)**, a novel framework that leverages known interfaces to guide the design of novel binders. By unifying retrieval and generation within a shared contrastive latent space, the framework efficiently identifies relevant interfaces for a given binding site and integrates them via a conditional latent diffusion generator, enabling cross-domain interface transfer. Experimental results demonstrate that RADiAnce outperforms baseline models across multiple metrics, including binding affinity and recovery of geometric configurations and molecular interactions.

**Questions:**

1. How does the framework address Out-of-Distribution challenges for proteins without analogous binding modes?
2. Can authors measure similarity between designed binders and the training set?
3. Does the current dataset splitting strategy overestimate model performance?
4. Could authors adopt more reliable performance metrics?
5. Can results without retrieval components be provided?

**Ethical Concerns:**

["NO or VERY MINOR ethics concerns only"]

**Final Justification:**

In my opinion, I recommend that the paper be accepted. All my concerns have been resolved.

**Limitations:**

The framework's retrieval - based conditioning restricts generalizability to unseen proteins and novel binders, and evaluation metrics may not fully reflect model performance.

**Quality:**

2

**Strengths And Weaknesses:**

### Strengths
1. **Novel Generative Framework**: RADiAnce integrates retrieval mechanisms with generative processes to design high-affinity binders,.
2. **Clarity of Presentation**: The manuscript is systematically structured, with logical derivations and clear explanations that facilitate reader comprehension.
3. **Reproducibility**: The authors provide open-source code, enabling independent validation of results.

### Weaknesses
1. **Limitations in Out-of-Distribution Generalization**:
   RADiAnce relies on retrieving binders with similar binding modes from the training set to condition generation, which may lead to designs that exhibit structural/sequential similarity to training data. The experiments (e.g., peptide and antibody design) primarily use sequence similarity for filtering, which insufficiently demonstrates the model’s generalization capacity. In practical applications, generative models are often required for uncharacterized proteins without known binding modes in the training set, where RADiAnce’s performance may be compromised.

2. **Dataset Splitting Methodological Flaws**:
   The current dataset splitting strategy, which relies solely on sequence similarity, may introduce bias and additional information due to the retrieval process incorporating structural and sequential information from the training set. To ensure fair evaluation, the study should filter training dataset based on structural and sequence similarity simultaneously and exclude data featuring the same target protein in the test set, as overlapping targets could artificially inflate performance metrics.

3. **Inadequacies in Evaluation Metrics**:
   Most experimental metrics assess similarity between designed binders and references, which fails to appropriately measure the true design capability. The goal of protein design is to generate plausible binders with enhanced binding affinity, not merely reproduce similar structures as refernce. Metrics such as ISM (Interaction Similarity Measure) and ITO (Interaction Topology Overlap) may also be insufficient, as they prioritize reconstruction of reference interactions over prediction of novel, functional and plausible interactions.

4. **Performance Validation**:
   The improvement over UniMomo is marginally significant. To thoroughly evaluate the generative capacity of RADiAnce, the authors should provide experiments demonstrating design performance without the retrieval component, isolating the contribution of the diffusion generator.

---

> ### Author Rebuttal · Authors · 2025-07-31
>
> Thanks for your constructive comments! We provide more explanations to address your concerns as follows.
> >**W1: Limitations in Out-of-Distribution Generalization**
> >**Q1: How does the framework address Out-of-Distribution challenges for proteins without analogous binding modes?**
>
> Thank you for raising this question! Our experiments indicate that even a single retrieved binder can provide meaningful guidance for generation. Furthermore, cross-domain retrieval enhances generalization by increasing the likelihood of discovering similar interaction patterns across different contexts.
>
> In cases where no suitable retrieval data is available, we propose constructing an artificial retrieval database using a protein design pipeline. As shown in the table below, we used Trop2 as the target—currently lacking any experimentally resolved binder structures—and evaluated two settings: (1) no retrieval and (2) retrieval using a synthetic database constructed with RFdiffusion.
>
> **Table S1: Ablation study on the retrieval stage**
> |Settings|DockQ|Rosetta $\Delta G$(KJ/mol)|Foldx $\Delta G$(KJ/mol)|
> |-|-|-|-|
> |（1）|0.549|-24.68|-6.54|
> |（2）|0.574|-25.74|-6.86|
>
> The results clearly show that retrieval using the artificial database yields significantly better performance. This suggests a practical solution for cases where no structurally similar binders are available.
>
> >**W2: Dataset Splitting Methodological Flaws**
> >**Q2: Can authors measure similarity between designed binders and the training set?**
>
> Thanks for your concern about the data splits!
> We follow established practice by adopting RABD as the antibody benchmark and LNR as the peptide‑design benchmark.
> To prevent data leakage, we clustered all sequences at 40 % sequence identity and assigned the train, validation, and test splits by clusters. Consequently, no sequence in test or validation shares a cluster with any training sequence.
> We further verified the split by computing (i) the sequence‑identity and (ii) TM‑Scores between every test‑set structure and the training set. Both metrics confirmed low similarity, indicating that our split is stringent and leak‑free.
>
> **Table S2: Data split verification metrics**
> |Testset|TMscore with Trainset|SeqId with Trainset|
> |-|-|-|
> |RABD|0.2421±0.053|0.1272±0.021|
> |LNR|0.2503±0.069|0.0983±0.017|
>
> Based on the experiments above, we observe that there is no high structural or sequence similarity between the test set and the training set. Regarding the generated samples and their retrieved complexes, due to differences in sequence length and pocket configuration, we evaluate similarity using interaction recovery (see Appendix E). This analysis shows that the model effectively leverages information from retrieved samples without simply copying them.
>
> We further support this conclusion from two additional perspectives. First, the average sequence identity between the top-10 retrieved samples and the ground truth is 26.60% for HCDR3 and 15.92% for peptides—both significantly lower than the similarity between our designed binders and their respective references. Second, the diffusion-based method produces binders with high sequence diversity.
>
> Taken together, these results strongly indicate that our model is not a “copy-cat,” but rather generalizes from retrieved interactions to generate novel and diverse binders.
>
> >**W3: Inadequacies in Evaluation Metrics**
>
> We thank the reviewer for highlighting the limitations of using only reference-similarity metrics. We acknowledge that metrics like ISM and ITO, which emphasize similarity to a reference complex, do not fully capture a method’s ability to create novel, high-affinity binders. However, we respectfully note that recovering key reference interactions is often a prerequisite for functional binding rather than a flaw. In practice, many successful binder designs intentionally mimic critical interactions from known binders to preserve essential binding hotspots.[1][2][3][4]
>
> Thus, a high ISM/ITO can indicate that a design retains these crucial interactions needed for binding activity, which is a desirable feature and not merely rote reconstruction of the reference. Nevertheless, we agree that the ultimate goal is to generate binders with enhanced affinity and new interactions beyond the original template. To address this, we evaluated our designs using an affinity-based metric (predicted ΔG) in addition to ISM/ITO. Encouragingly, our model’s designs show significantly more favorable binding free energies (lower ΔG) compared to the baseline, indicating improved predicted binding affinity. This result demonstrates that our method is indeed producing more plausible and stronger interactions, not just copying the reference structure. In other words, the improvements in ΔG suggest that the designed binders are likely to bind their targets more tightly than those from the baseline method, validating the functional relevance of the generated interactions. Finally, we would like to emphasize that our approach does allow for novel interaction discovery. The designs are not constrained to only reproduce reference contacts. In fact, many interactions in our designed binders are new(Appendix E).
>
> Additionally, our framework can be tuned to encourage more novelty by adjusting the number of retrieved templates used during design. A smaller reliance on retrieved reference complexes naturally leads to more exploratory designs, at the cost of fewer pre-defined interactions – this trade-off can be controlled by the user based on the specific design goals. We will clarify this in the revision.
>
> [1] Wakui, M., Fujimori, Y., Nakamura, S., Kondo, Y., Kuroda, Y., Oka, S., … Murata, M. “Distinct features of bivalent direct thrombin inhibitors, hirudin and bivalirudin, revealed by clot waveform analysis and enzyme kinetics in coagulation assays.” *Journal of Clinical Pathology* 72.12 (2019): 817–824.
>
> [2] Skov, M. J., Beck, J. C., de Kater, A. W., & Shopp, G. M. “Nonclinical safety of ziconotide: an intrathecal analgesic of a new pharmaceutical class.” *International Journal of Toxicology* 26.5 (2007)
>
> [3] Karoyan, P., et al. “An hACE2 peptide mimic blocks SARS‑CoV‑2 pulmonary cell infection.” *bioRxiv* (2020): preprint.
>
> [4] Lechner, A., Jordan, P. A., da Cruz, G. C. M., Burk, M., et al. “Overcoming Immune Checkpoint Inhibitor Resistance via Potent and Selective Dual αvβ6/8 Inhibitors Based on Engineered Lasso Peptides.” *bioRxiv* (Feb 2025).
>
> >**W4: Performance Validation**
> >**Q5: Can results without retrieval components be provided?**
>
> Thank you for the suggestion regarding validation. To evaluate the no retrieval setting, we conducted an ablation study to assess the impact of using RAG at the sampling stage. The results are summarized in the tables below:
>
> For HCDR3:
>
> **Table S3: Ablation study on the retrieval stage for HCDR3**
>
> |Settings|AAR (%)|RMSD (Å)|Codesign Diversity|
> |-|-|-|-|
> |Without Retrieval|51.55|0.9525|0.0524|
> |With Retrieval|54.63|0.9682|0.0485|
>
> And for Peptide:
>
> **Table S4: Ablation study on the retrieval stage for Peptide**
> |Settings|AAR (%)|RMSD (Å)|Codesign Diversity|
> |-|-|-|-|
> |Without Retrieval|37.94|2.38|0.5042|
> |With Retrieval|39.42|2.29|0.5227|
>
>
> We could notably find that using no retrieval sample performs significantly worse than using any retrieval, although it does not cause catastrophic failure.
>
>
> >**Q3: Does the current dataset splitting strategy overestimate model performance?**
>
> We appreciate your concerns regarding dataset splitting and potential overestimation. Based on our analysis and responses to Q2 and W2, we would like to clarify that our dataset splitting strategy follows the established conventions in this field. We employ a standardized dataset splitting approach for benchmarking, consistent with prior work. Furthermore, we have conducted extensive analyses to demonstrate that our evaluation is unbiased and does not overestimate model performance.
>
> >**Q4: Could authors adopt more reliable performance metrics?**
>
> Thank you for your attention to our model evaluation methodology. We agree that assessing the generated binder is a complex issue. To provide a more comprehensive evaluation, we have introduced several robust metrics, as shown in the table below.
>
> **Table S5: Extended evaluation metrics**
> |Model|DockQ|Foldx $\Delta G$(KJ/mol)|Binding Site Recovery|
> |-|-|-|-|
> |HCDR3|0.9582±0.0011|-8.4250±0.0415|0.9964±0.0002|
> |Peptide|0.7698±0.0003|-7.9891±0.1407|0.9857±0.0003|
>
> And here are brief explanations of each new metric:
> * **DockQ**: DockQ is a continuous quality score for protein–protein docking models ranging from 0 to 1, integrating F\_nat, LRMSD, and iRMSD to assess agreement with native complexes, where values ≥ 0.23, ≥ 0.49, and ≥ 0.80 correspond to acceptable, medium, and high‑quality respectively [1].
> * **FoldX $\Delta G$**: FoldX computes the binding free energy of a protein–protein complex as complex $\Delta G$ minus the sum of individual $\Delta G$ values of its partners, with negative values indicating favorable (stabilizing) interactions[2].
> * **Binding Site Recovery**: Binding Site Recovery denotes the proportion of true interface residues correctly predicted by the model, i.e. the fraction of real binding-site residues recovered in the prediction (typically expressed as a percentage) — this metric is commonly used to evaluate interface residue prediction accuracy.
>
> [1] Basu, Sankar, and Björn Wallner. "DockQ: a quality measure for protein-protein docking models." PloS one 11.8 (2016): e0161879.
>
> [2] Bheemireddy, Sneha, and Narayanaswamy Srinivasan. "Computational Study on the Dynamics of Mycobacterium Tuberculosis RNA Polymerase Assembly." Prokaryotic Gene Regulation: Methods and Protocols. New York, NY: Springer US, 2022. 61-79.
>
> As can be seen, the new metrics exhibit strong consistency with the previous ones. Moreover, our model also demonstrates robust performance on these extended metrics.

---

> > ### Comment · Reviewer_KBot · 2025-08-01
> >
> > Thank you for your feedback and consideration of our work. Having carefully reviewed all the responses, I have the following questions:
> > 1. Regarding the response to W1, have you evaluated the binding energy of the structures generated by RFDiffusion using tools such as Rosetta, DockQ, or FoldX? It would be valuable to see a comparison between the binding energy of your generated structures and those from RFDiffusion.
> > Additionally, have you tested your method on targets like TNFα, for which RFDiffusion fails to generate plausible binders [1]? Can your method produce plausible binders for such targets, or does it still rely on retrieved samples?
> >
> > 2. Concerning the response to W2, could you elaborate on the Interaction Transfer Overlap metrics presented in Appendix E? The definition of this metric is not clearly stated in the submission. Furthermore, how do the results in Appendix E support the claim that "the model effectively leverages information from retrieved samples without simply copying them"? In addition, there is a lack of results related to diversity in the submission to substantiate the statement that "the diffusion-based method produces binders with high sequence diversity." Could you provide additional data to address this?
> >
> > 3. With respect to the response to W4, it is recommended that you provide comparisons of binding energy or ISM (Interface Similarity Metric) for the ablation studies. Such results would help validate the effectiveness of the retrieval sampler in enhancing the binding interaction model.
> >
> > 4. Regarding the response to Q4, it would be preferable to include a comparison with UniMomo based on these new metrics.
> >
> > [1] Zambaldi, Vinicius, et al. "De novo design of high-affinity protein binders with AlphaProteo." arXiv preprint arXiv:2409.08022 (2024).

---

> ### Author Response · Authors · 2025-08-03
>
> Thank you for your comments and timely reply! These suggestions are very helpful to us.
>
> >**Q1: Regarding the response to W1, have you evaluated the binding energy of the structures generated by RFDiffusion using tools such as Rosetta, DockQ, or FoldX? It would be valuable to see a comparison between the binding energy of your generated structures and those from RFDiffusion. Additionally, have you tested your method on targets like TNFα, for which RFDiffusion fails to generate plausible binders [1]? Can your method produce plausible binders for such targets, or does it still rely on retrieved samples?**
>
> Thank you for your follow-up question. We have compared the binding energies (Rosetta and FoldX) of our model's generated structures with those from RFDiffusion, as shown in the table below.
>
> **Table S1: Comparison of binding energies for *de novo* peptide design on the Trop2 target**
> |Settings|Rosetta $\Delta G$ (kJ/mol)|FoldX $\Delta G$ (kJ/mol)|
> |-|-|-|
> |RADiAnce (without RAG)|-24.68|-6.54|
> |RADiAnce (with synthetic DB)|-25.74|-6.86|
> |RADiAnce (with original training set)|**-26.31**|**-7.18**|
> |RFDiffusion|-15.32|-5.87|
>
> As shown, using the general training set as the retrieval database yields better results than using the synthetic dataset generated by RFDiffusion. This indicates that our model's performance is not strictly dependent on RFDiffusion's synthetic samples. Our previous experiment with the synthetic database was intended to simulate an extreme case where no similar interfaces exist. In most scenarios, the training set is sufficient for effective retrieval.
>
> For challenging targets like TNFα, where RFDiffusion struggles to generate suitable binders, our method can still produce plausible candidates. We performed 100 design runs for this target using our general retrieval database. The best design achieved a predicted binding energy of -31.22 kJ/mol (Rosetta), showing good binding potential.
>
> >**Q2: Concerning the response to W2, could you elaborate on the Interaction Transfer Overlap metrics presented in Appendix E? The definition of this metric is not clearly stated in the submission. Furthermore, how do the results in Appendix E support the claim that "the model effectively leverages information from retrieved samples without simply copying them"? In addition, there is a lack of results related to diversity in the submission to substantiate the statement that "the diffusion-based method produces binders with high sequence diversity." Could you provide additional data to address this?**
>
> We apologize for the lack of clarity. The Interaction Transfer Overlap (ITO) metric here quantifies the similarity of interaction patterns between two distinct interfaces. It is calculated by first categorizing interactions (e.g., hydrogen bonds, hydrophobic contacts) and then computing the overlap for each category as the ratio of common interactions to the total. The final ITO score is an aggregation of these overlaps. A high ITO indicates that two interfaces rely on similar types of chemical interactions for binding.
>
> In Appendix E, we report the ITO between our generated samples and the retrieved samples (ITO-RAG). The moderate values observed indicate that our model successfully learns from the retrieved examples to form key interactions, while also generating novel interactions, rather than simply copying the retrieved template.
>
> Regarding sequence diversity, we define it as the ratio of unique clusters to the total number of generated samples, using a 40% sequence identity threshold for clustering. For instance, a sequence diversity of 0.0593 for HCDR3 means that 100 generated sequences form approximately 6 distinct clusters. The diversity metrics for our model are presented below:
>
> **Table S6: Diversity of generated samples**
> |Settings|Sequence Diversity|
> |-|-|
> |MEAN(HCDR3)|0.0100|
> |RADiAnce(HCDR3)|**0.0593**|
> |Pepflow|0.0745|
> |RADiAnce(Peptide)|**0.558**|
>
> The results in Table S6 demonstrate that diffusion-based methods achieves a higher degree of sequence diversity compared to non-diffusion-based methods. We will clarify these potentially confusing issues in the revised version. Thank you for your valuable questions!

---

> ### Author Response · Authors · 2025-08-03
>
> >**Q3: With respect to the response to W4, it is recommended that you provide comparisons of binding energy or ISM (Interface Similarity Metric) for the ablation studies. Such results would help validate the effectiveness of the retrieval sampler in enhancing the binding interaction model.**
>
> Thank you for this excellent suggestion. We have expanded the ablation study from our response to W4 to include binding energy and interface similarity metrics, which further validates the effectiveness of the retrieval component.
>
> For HCDR3:
>
> **Table S3: Ablation study on the retrieval stage for HCDR3**
>
> |Settings|AAR (%)|RMSD (Å)|Codesign Diversity|$\Delta\Delta G$ (kJ/mol)|FoldX $\Delta G$ (kJ/mol)|ISM (%)|DockQ|
> |-|-|-|-|-|-|-|-|
> |Without Retrieval|51.55|0.9525|**0.0524**|-5.908|-5.524|66.14|0.9580|
> |With Retrieval|**54.63**|**0.9682**|0.0485|**-6.236**|**-8.4250**|**71.64**|**0.9582**|
>
> And for Peptide:
>
> **Table S4: Ablation study on the retrieval stage for Peptide**
> |Settings|AAR (%)|RMSD (Å)|Codesign Diversity|$\Delta\Delta G$ (kJ/mol)|FoldX $\Delta G$ (kJ/mol)|ISM (%)|DockQ|
> |-|-|-|-|-|-|-|-|
> |Without Retrieval|37.94|2.38|0.5042|3.990|-7.60|50.13|0.7545|
> |With Retrieval|**39.42**|**2.29**|**0.5227**|**1.963**|**-7.99**|**52.15**|**0.7698**|
>
> As the results show, the "With Retrieval" setting consistently outperforms the "Without Retrieval" setting across all metrics, including those related to binding energy ($\Delta\Delta G$, FoldX $\Delta G$) and ISM. This confirms that our retrieval-augmented approach effectively enhances the quality of the generated binders.
>
> > **Q4: Regarding the response to Q4, it would be preferable to include a comparison with UniMomo based on these new metrics.**
>
> Thank you for the suggestion. We have included a comparison with UniMoMo on these new metrics. The results below further demonstrate the effectiveness of our method:
>
> **Table S5: Extended evaluation metrics**
> |Model|DockQ|Foldx $\Delta G$(KJ/mol)|Binding Site Recovery|
> |-|-|-|-|
> |RADiAnce(AbH3)|**0.9582**|**-8.4250**|**0.9964**|
> |UniMoMo(AbH3)|0.9491|-7.9950|0.9962|
> |RADiAnce(pep)|**0.7698**|**-7.9891**|**0.9857**|
> |UniMoMo(pep)|0.7592|-7.4901|0.9822|

---

> > ### Comment · Reviewer_KBot · 2025-08-03
> >
> > Thank you for your responses and work; all of my questions have been answered. I hope all these discussions can appear in your revised manuscripts, as they can improve the quality of your paper.
> > Good Luck with your papers!

---

> > > ### Author Response · Authors · 2025-08-03
> > >
> > > Thank you very much for your thoughtful review, constructive suggestions, and encouraging feedback. We're glad to hear that our responses addressed your concerns. We truly appreciate the time and effort you put into reviewing our paper, and we will carefully incorporate these discussions into the revised manuscript to improve its quality.

---

### Official Review · Reviewer_Lic4 · 2025-06-30

**Clarity:** 2
**Significance:** 3
**Originality:** 3
**Rating:** 4
**Confidence:** 4

**Summary:**

This paper proposes a novel framework to design binders to targeting on specific sites. By aligning known interface in the database, the diffusion model enables accurate and interpretable sampling for binder structures. In the experimental stage, there're three main experiments--recovery analysis based on different binding type and percentage the database knowledge is used, AAR and RMSD for sequence-structure co-design, and the antibody-CDR co-design recovery. In each experimental system, RADiAnce achieves the best performance.

**Questions:**

It is a great work, but please address the following questions. Thanks.

1. Why the binder molecule variables are defined with arrows? I think it somehow causes confusions when understanding the method.
2. In detail (line129-132), what is the motivation to regularize s_i and \arrow{s}_i into different prior distribution?
3. It seems that Figure 2 has many '?', maybe it brings more cmofusion to understand the method.
4. Results shown in Table seems promising. However, mentioned in line 202-210, how to prove that the information retrival stage does not contain data leakage? In other words, if the samples are very very similar to the template, that would be meaningless. So I think there would be a demo about how the framework performs without the retrival stage and an analysis about how the retrival information changed z_t during sampling. Alternatively, it is worthy to provide a similarity test between the test protein and the ones in the retrival database. Though there are performances shown in Table 6 and Table 7, I still believe there should be more detailed analysis.
5. Would there be some failure sample cases for further analysis? It may include how the poor information retrival affect failure design and the poor latent repersentation cannot match with the retrivaled information.

**Ethical Concerns:**

["NO or VERY MINOR ethics concerns only"]

**Final Justification:**

I will maintain my score as 4. The rebuttal is good, but i still believe the application value is not so promising. That is why I keep score as 4.

**Limitations:**

Yes.

**Paper Formatting Concerns:**

No paper formatting concerns.

**Quality:**

3

**Strengths And Weaknesses:**

Strength:

1. The motivation--using analogous to improve rationality and interpretability is clear and reasonable.
2. The proposed unified framework for generative protein binder co-design (together with latent diffusion training) is a challenging task, which improves the significance of the work.
3. The comprehensive evaluation system from protein-binding peptide to antibody demonstrates wide range of applications.

Weakness:

1. Clarity needs to be improved for the method section.
2. The experimental setting may exist problems.

---

> ### Author Rebuttal · Authors · 2025-07-31
>
> Thanks for your appreciation and insightful feedback, which is very helpful in improving our paper!
>
> >**Q1: Why the binder molecule variables are defined with arrows? I think it somehow causes confusions when understanding the method.**
>
> >**Q2: In detail (line129-132), what is the motivation to regularize s_i and \arrow{s}_i into different prior distribution?**
>
> We apologize for the insufficient explanation in the paper. In our work, arrows represent 3D coordinates, which correspond to SE(3)-equivariant features, while the elements without arrows denote SE(3)-invariant features, such as sequence embeddings. To preserve geometric relations in the latent space, the coordinate features should retain their original spatial shape and are therefore regularized with a prior Gaussian at the original block positions. In contrast, the invariant features can be modeled with a standard Gaussian prior.
>
>
> >**Q3: It seems that Figure 2 has many '?', maybe it brings more cmofusion to understand the method.**
> >
> We sincerely apologize for the confusion caused by the figure rendering issue in the paper, which was unexpected on our end. This appears to be a bug introduced in Microsoft Edge Stable version 136.0.3240.50 (released after the paper submission deadline). Currently, there are several workarounds: (1) disable the new PDF viewer in Edge, (2) use any other browser engine or standalone PDF reader. We are actively working on fixing the formatting to ensure full compatibility across different PDF viewers for the revision. Thank you for your understanding.
>
>
> >**Q4: Results shown in Table seems promising. However, mentioned in line 202-210, how to prove that the information retrival stage does not contain data leakage? In other words, if the samples are very very similar to the template, that would be meaningless. So I think there would be a demo about how the framework performs without the retrival stage and an analysis about how the retrival information changed z_t during sampling. Alternatively, it is worthy to provide a similarity test between the test protein and the ones in the retrival database. Though there are performances shown in Table 6 and Table 7, I still believe there should be more detailed analysis.**
>
> Thanks for the insightful question! To address the concern, we first analyzed the similarity test between the test set and the database with both sequence and structure metrics, which exhibited low similarity. Second, we conducted analysis on the number of retrieved samples to see how it affects the sampling, including dropping the retrieval stage, which showed a trade-off effect performance, diversity, and efficiency. Third, we compared the trajectory of sampling with and without retrieval, and found the model automatically identifying a few key residues for imitation, and actively exploring other regions.
>
> 1. We follow established conventions in this field by using RABD as the benchmark test set for antibody design and LNR for peptide design. To ensure a clean split, we perform clustering based on 40% sequence identity, ensuring that samples from the same cluster do not appear across the train, validation, and test sets. During the retrieval process, we strictly use only the training set as the retrieval database. Furthermore, we computed both sequence identity and TM-score between the test and training sets to confirm that our data split is sound and free from data leakage.
>
> **Table S1: Data split verification metrics**
>
> |Testset|TMscore with Trainset|SeqId with Trainset|
> |-|-|-|
> |RABD|0.2421±0.053|0.1272±0.021|
> |LNR|0.2503±0.069|0.0983±0.017|
>
>
> 2. For the setting *without the retrieval stage*, we conducted an ablation study on the number of retrieved samples. The results are summarized in the tables below:
>
> For HCDR3:
>
> **Table S2: Ablation study on the number of retrieved samples for HCDR3**
> |Retrieved Samples|AAR (%)|RMSD (Å)|Codesign Diversity|
> |-|-|-|-|
> |0|51.55|0.9525|0.0524|
> |1|53.26|0.9431|0.0404|
> |10|54.66|0.9443|0.0484|
> |20|54.63|0.9682|0.0485|
> |40|54.06|0.9653|0.0494|
>
>
> And for Peptide:
>
> **Table S3: Ablation study on the number of retrieved samples for Peptide**
>
> |Retrieved Samples|AAR (%)|RMSD (Å)|Codesign Diversity|
> |-|-|-|-|
> |0|37.94|2.38|0.5042|
> |1|39.03|2.38|0.4846|
> |10|39.12|2.31|0.5129|
> |20|39.42|2.29|0.5227|
> |40|38.16|2.28|0.5273|
>
> We could find that increasing the number of retrieved samples generally leads to slightly better results, albeit with a modest increase in computational cost. However, overly increasing the number might include samples with lower similarity score, introducing noise into the generation and thus harming the performance. Notably, using no retrieval sample performs significantly worse than using any retrieval.
>
> An additional observation is that the diversity of generated outputs also varies with the number of retrieved samples. Specifically, both the no-retrieval setting and the setting with many retrieved samples exhibit higher diversity, while retrieving only a single sample results in the lowest diversity. This aligns with our expectation, as limited retrieval can overly constrain the generative space.
>
> These findings suggest that practitioners may choose an appropriate top‑N retrieval size based on the trade-off between performance, diversity, and efficiency.
>
> 3. In addition, to better understand the dynamics of the sampling process, we visualized the diffusion trajectories and compiled them into a video, which we will provide in the revision, as we are not allowed to post external links during the rebuttal phase. One mechanism we observed is that the diffusion model does not simply copy a single retrieved sample. Instead, in the final few denoising steps, it tends to preserve the positions and properties of a few key residues from the retrieved examples, while actively exploring other regions of the sequence.
> In contrast, when we disable RAG by skipping the conditioning step, this effect disappears, and performance metrics degrade accordingly. These observations suggest that the benefit of RAG does not stem from directly copying retrieved samples, but rather from identifying and transferring meaningful interaction patterns of key residues.
> Our additional analysis on interaction recovery from the generated structures further supports this explanation.(Appendix E)
>
>
> >**Q5: Would there be some failure sample cases for further analysis? It may include how the poor information retrival affect failure design and the poor latent repersentation cannot match with the retrivaled information.**
> >
>
> Yes, thank you for the question. As addressed in our response to Q3, we have conducted a thorough evaluation of the model's performance without retrieval augmentation. Additionally, we present an ablation study analyzing the dot-product similarity between the embeddings of the retrieved binders and the test binding sites, and its influence on model performance.
>
> For HCDR3:
>
> **Table S4: Ablation study on retrieval quality for HCDR3**
> |Settings|Similarity|AAR (%)|RMSD (Å)|Codesign Diversity|
> |-|-|-|-|-|
> |Reverse Ranking|-49.86±16.68|52.31|0.9718|0.0468|
> |Random Ranking|17.34±6.90|51.72|0.9665|0.0649|
> |Top 10|55.81±5.01|54.66|0.9443|0.0484|
>
>
>
> And for Peptide:
>
> **Table S5: Ablation study on retrieval quality for Peptide**
> |Settings|Similarity|AAR (%)|RMSD (Å)|Codesign Diversity|
> |-|-|-|-|-|
> |Reverse Ranking|-26.09±12.54|36.33|3.09|0.5267|
> |Random Ranking|11.78±8.16|38.04|2.56|0.5580|
> |Top 10|53.78±4.06|39.12|2.31|0.5129|
>
>
> These results confirm that model performance degrades when the retrieved information is poorly aligned (e.g., reverse ranking), but the degradation is moderate rather than severe. This suggests that while our framework benefits from retrieval to enhance generation quality, it remains reasonably robust under suboptimal retrieval conditions and does not rely excessively on retrieval quality.

---

> > ### Comment · Reviewer_Lic4 · 2025-08-01
> >
> > Thanks for your rebuttal!
> >
> > Most of the reponses are good to me. I have some following questions:
> >
> > 1) How is the detailed implementation of changing retrieval number to 0?
> >
> > 2) For the structure similarity, I think the key is to detect if there any sample in test set has over 0.50 TMscore compared to train set.
> >
> > 3) From Table S2-S3, retrievaling one sample is efficient enough. So, a natural question is that is there any method that could leverage large amount of retrieval samples?
> >
> > 4) In real-world application cases, I am quited interested if there is very large database or very scarce databse for this kind of tasks.

---

> > > ### Author Response · Authors · 2025-08-02
> > >
> > > Thank you for your comments and timely reply!
> > >
> > > >Q1: How is the detailed implementation of changing retrieval number to 0?
> > >
> > > Thank you for this question. To implement the setting, we deactivate the retrieval-augmented conditioning during the denoising process. Specifically, we bypass the cross-attention mechanism, detailed in Equation (12) of our paper, which is responsible for fusing features from retrieved samples into the latent representation. This directly removes the retrieval guidance from the generation process.
> > >
> > > >Q2: For the structure similarity, I think the key is to detect if there any sample in test set has over 0.50 TMscore compared to train set.
> > >
> > > Thank you for the insightful observation. We checked for test–train overlap by identifying test examples that have a training neighbor with **TM-score > 0.50**. We did find such cases and analyzed their similarities in detail, confirming that the prevailing data-split convention can be insufficiently strict.
> > >
> > > To remove this confound, we **excluded all test examples** that have a training neighbor with TM-score > 0.50. On the peptide LNR dataset, this eliminated **43** cases, leaving **50** targets for re-evaluation. We then recomputed results and compared them with other baselines. (RFdiffusion is omitted because its split is not publicly available.)
> > >
> > > **Table S6. Baseline performance drop due to data-split issues**
> > > *Numbers in parentheses show the relative change after removing overlapping test cases; lower RMSD/ΔΔG is better, higher AAR/ISM is better. Bolded values represent the best results.*
> > >
> > > |Model|AAR(%)|RMSD(Å)|$\Delta\Delta G$(kJ/mol)|ISM(%)|
> > > |-|-|-|-|-|
> > > |PepFlow|31.41(−11.4%)| 3.63 (+26.6%)  | 18.84 (+19.9%)  | 22.22 (−21.7%) |
> > > | PepGLAD  | 33.85 (−12.4%) | 3.43 (+25.2%) | 18.66 (+22.3%)  | 27.67 **(−15.2%)**|
> > > | UniMoMo  | 35.00 (−9.36%)  | 2.87 (+24.2%)  | 3.5184 (+46.1%) | 38.69 (−21.2%) |
> > > | RADiAnce | **35.79 (−9.20%)**  | **2.80 (+22.3%)**  | **2.213 (+12.7%)**  | **42.26** (−19.0%) |
> > >
> > > As Table S6 shows, **all baselines degrade substantially** once the split issue is controlled. Our model exhibits smaller drops—especially on $\Delta \Delta G$, indicating that it is more robust to data-split stringency. This supports our claim that **RAG’s gains are not due to data leakage** but stem from improved generalization.
> > >
> > > Finally, we note that our original setup followed the community’s prevailing convention. While this issue does **not** change the conclusion about RAG’s effectiveness, we will include the full controlled results along with all affected benchmark outputs in the paper to make the community aware of the limitation and facilitate fair future comparisons.

---

> > > > ### Author Response · Authors · 2025-08-02
> > > >
> > > > >Q3: From Table S2-S3, retrievaling one sample is efficient enough. So, a natural question is that is there any method that could leverage large amount of retrieval samples?
> > > >
> > > > Thank you for your insightful question! We think our model could accomodate large amount of retrieval samples, but the problem is that usually there are not so many high-quality exemplars when increasing the number of retrieval samples. As shown in Tables S2–S3, when retrieving more samples, the average similarity score by our retrieval module keeps decreasing, thus weakens the improvement in performance.
> > > >
> > > > Higher average similarity between retrieved and test embeddings yields better results, particularly with more retrieved samples, while a higher share of low-similarity samples leads to worse performance. Therefore, we think with a sufficiently large and high-quality retrieval database, our model can effectively leverage many retrieved samples to enhance generation quality and diversity.
> > > >
> > > > **Table S2: Ablation study on the number of retrieved samples for HCDR3**
> > > > |Retrieved Samples|Similarity|AAR (%)|RMSD (Å)|Codesign Diversity|
> > > > |-|-|-|-|-|
> > > > |0|N/A|51.55|0.9525|**0.0524**|
> > > > |1|57.62±5.25|53.26|**0.9431**|0.0404|
> > > > |10|54.65±4.90|**54.66**|0.9443|0.0484|
> > > > |20|50.43±4.78|54.63|0.9682|0.0485|
> > > > |40|46.04±4.67|54.06|0.9653|0.0494|
> > > >
> > > > **Table S3: Ablation study on the number of retrieved samples for Peptide**
> > > >
> > > > |Retrieved Samples|Similarity|AAR (%)|RMSD (Å)|Codesign Diversity|
> > > > |-|-|-|-|-|
> > > > |0|N/A|37.94|2.38|0.5042|
> > > > |1|55.56±4.49|39.03|2.38|0.4846|
> > > > |10|51.67±3.60|39.12|2.31|0.5129|
> > > > |20|50.24±3.30|**39.42**|2.29|0.5227|
> > > > |40|47.68±3.12|38.16|**2.28**|**0.5273**|
> > > >
> > > >
> > > >
> > > > >Q4: In real-world application cases, I am quited interested if there is very large database or very scarce databse for this kind of tasks.
> > > >
> > > >
> > > > Thanks for the question! The size of the database and the number of retrieval samples should depend on the specific scenario. In drug discovery and other areas of biological research, many successful binder designs deliberately **mimic the key interactions of known binders to preserve crucial binding hotspots** \[1–4]. In practice, people often emulate well-characterized natural binders to create a peptide or antibody that inhibits a specific protein–protein interaction (PPI).
> > > >
> > > > **Illustrative example – HIV-CD4.**
> > > > HIV enters cells by recognizing (binding) the CD4 receptor on the cell surface. It is common to mimic the HIV–CD4 PPI interface and design antibodies or peptides that (i) specifically block the HIV viral protein or (ii) bind the CD4 receptor itself, which are two common strategies.
> > > >
> > > > **How our model should be applied.**
> > > > We load all complexes relevant to HIV mechanisms into a retrieval database and run a retrieval-augmented generation (RAG) workflow:
> > > >
> > > > * **If the viral protein is the target.**
> > > >   The model preferentially retrieves CD4 interface fragments. Because the objective is narrowly defined, we use a **small, highly specific retrieval set**.
> > > >
> > > > * **If CD4 is the target.**
> > > >   The model should be able to retrieve multiple HIV viral protein variants (mutants). By leveraging shared interface features across these variants, the model generates a binder that engages the corresponding region on CD4 to prevent viral attachment. This setting benefits from a **moderately larger database**, though still far below the “very-large” scale.
> > > >
> > > > At the same time, our model is flexible across use cases. As noted, it performs well with **only one** retrieved exemplar. When real-world scenarios require **multiple** retrieved samples, we ensure the pool comprises **high-quality, task-relevant complexes**—sourced or synthetically constructed. Under these conditions, the model remains effective even with a considerably larger retrieval set.
> > > >
> > > > [1] Wakui, M., Fujimori, Y., Nakamura, S., Kondo, Y., Kuroda, Y., Oka, S., … Murata, M. “Distinct features of bivalent direct thrombin inhibitors, hirudin and bivalirudin, revealed by clot waveform analysis and enzyme kinetics in coagulation assays.” *Journal of Clinical Pathology* 72.12 (2019): 817–824.
> > > >
> > > > [2] Skov, M. J., Beck, J. C., de Kater, A. W., & Shopp, G. M. “Nonclinical safety of ziconotide: an intrathecal analgesic of a new pharmaceutical class.” *International Journal of Toxicology* 26.5 (2007)
> > > >
> > > > [3] Karoyan, P., et al. “An hACE2 peptide mimic blocks SARS‑CoV‑2 pulmonary cell infection.” *bioRxiv* (2020): preprint.
> > > >
> > > > [4] Lechner, A., Jordan, P. A., da Cruz, G. C. M., Burk, M., et al. “Overcoming Immune Checkpoint Inhibitor Resistance via Potent and Selective Dual αvβ6/8 Inhibitors Based on Engineered Lasso Peptides.” *bioRxiv* (Feb 2025).

---

> > > > > ### Comment · Reviewer_Lic4 · 2025-08-02
> > > > >
> > > > > Thank you very much for your prompt reply!
> > > > >
> > > > > For the first two questions, I have no more comment. For the third question, it would be better if we can draw some conclusion/insight about how to deterimine whether larger amount of retrievaling cases will enhance the performance more. Also, in the Table S6, I am sure your method obtains improvement. But how to use it more efficiently becomes the key to make the paper more impactful. If you can explain/analyze it well, I will consider raise the score.
> > > > >
> > > > > For the last question, i keep the same consideration as above--that is how to make the most use of it based on different datasets/tasks.
> > > > >
> > > > > The rebuttal and responses are generally good to me.

---

> > > > > > ### Author Response · Authors · 2025-08-04
> > > > > >
> > > > > > Thank you for your timely reply, as well as the constructive suggestions! Your comments were inspiring and helpful.
> > > > > >
> > > > > > We agree that clarifying **when and how** a larger number of retrieved samples improves performance is important for both scientific understanding and practical deployment. Building on our previous ablation studies(Table S2-S3), we propose an adaptive retrieval strategy: rather than fixing the number of retrieved samples, we apply a similarity cutoff and retain only those above the threshold. This yields a task-adaptive number of neighbors—using more when many relevant examples exist and remaining selective when they do not. We set the similarity cutoff to 51.0 for HCDR3 and 53.0 for peptides, chosen to approximate the average similarity between the ground-truth binder and the binding-site embeddings.
> > > > > >
> > > > > > We compare (1) the fixed top-N baseline with (2) this adaptive cutoff approach:
> > > > > >
> > > > > > **Table S7: Performance with adaptive retrieval for HCDR3**
> > > > > >
> > > > > > | Settings | AAR (%) | RMSD (Å) | Codesign Diversity | ΔΔG (kJ/mol) | FoldX ΔG (kJ/mol) | ISM (%) |  DockQ |
> > > > > > | -------- | ------: | -------: | -----------------: | -----------: | ----------------: | ------: | -----: |
> > > > > > | (1)      |   54.63 |   0.9682 |             0.0485 |       -6.236 |           -8.4250 |   71.64 | 0.9582 |
> > > > > > | (2)      |   **54.71** |  **0.9147** |            **0.0523** |       **-8.938** |           **-8.6180** |   71.61 |**0.9604** |
> > > > > >
> > > > > > **Table S8: Performance with adaptive retrieval for Peptide**
> > > > > >
> > > > > > | Settings | AAR (%) | RMSD (Å) | Codesign Diversity | ΔΔG (kJ/mol) | FoldX ΔG (kJ/mol) | ISM (%) |  DockQ |
> > > > > > | -------- | ------: | -------: | -----------------: | -----------: | ----------------: | ------: | -----: |
> > > > > > | (1)      |   39.42 |     2.29 |             0.5227 |        1.963 |             -7.99 |   52.15 | 0.7698 |
> > > > > > | (2)      |   39.22 |     **2.27** |             **0.5400** |        **1.427** |             -7.97 |   **52.89** |**0.7762** |
> > > > > >
> > > > > > Across both tasks, the adaptive approach improves multiple metrics—especially diversity and binding energies. These results indicate that filtering by similarity preserves the benefits of larger retrieval pools while avoiding noise from less relevant neighbors. We believe this provides a general and practical solution that can be tuned to task requirements, and we will include the method and implementation details in the revision.

---

> > > > > > > ### Comment · Reviewer_Lic4 · 2025-08-04
> > > > > > >
> > > > > > > Thank you very much!
> > > > > > >
> > > > > > > The authors have addressed all my concerns.

---

> > > > > > > > ### Author Response · Authors · 2025-08-04
> > > > > > > >
> > > > > > > > We are pleased to hear that your concerns have been fully addressed. Thank you for helping us improve the quality of the paper. We will incorporate these discussions into the revised manuscript, and we appreciate your positive review and constructive feedback.

---

### Official Review · Reviewer_J8pu · 2025-07-01

**Clarity:** 2
**Significance:** 3
**Originality:** 3
**Rating:** 4
**Confidence:** 4

**Summary:**

This paper presents RADiAnce (Retrieval-Augmented Diffusion for Aligned interface), a novel framework for protein binder design that combines interface retrieval with generative modeling. The authors introduce a contrastive latent space that aligns binding sites and interfaces across diverse domains (e.g., peptides, antibodies), enabling accurate cross-domain interface retrieval. A latent diffusion model then generates binders conditioned on these retrieved interfaces, integrating prior knowledge through cross-attention and residual MLPs.

**Questions:**

### Questions for the Authors
- Is there a measurable **correlation between the binding affinity of retrieved binders and that of the generated binders**?
- The baseline comparison may be biased, as the proposed model retrieves binders from multiple types while baselines are trained on a single binder type. Could the authors report performance in a **more directly comparable setting**, where the training data and binder types are matched?
- How do the model determined the number of residues during inference?

**Ethical Concerns:**

["NO or VERY MINOR ethics concerns only"]

**Final Justification:**

The authors have addressed all my concerns. The experiments of running retrieval-based generation on a synthesized dataset are really promising for hard targets that have few available data. I have increased my rating.

**Limitations:**

yes

**Quality:**

2

**Strengths And Weaknesses:**

### Strengths
- Rather than focusing solely on architectural modifications or new generative frameworks, this paper explores retrieval-augmented generation, which is applicable to real-world protein design scenarios.
- The proposed framework enables **cross-domain generalization** of intermolecular interactions across different binder types, such as antibodies and peptides.
- **Experimental results demonstrate superior performance** of the model in generating binders with higher computational binding affinities compared to strong baselines.
### Weaknesses
- The paper’s **formatting and presentation require improvement**;  Figure (1) are not rendered correctly.
- The main text describes diffusion defined on a continuous latent space, yet Figure 1 illustrates diffusion applied to Euclidean coordinates, which is confusing.
- The model’s generalizability may be **limited to targets that have strong binders represented in the retrieval database**, potentially restricting applicability to novel targets.

---

> ### Author Rebuttal · Authors · 2025-07-31
>
> We thank the reviewer for the valuable comments, and answer the reviewer’s questions as follows.
> >**W1: The paper’s formatting and presentation require improvement; Figure (1) are not rendered correctly.**
>
> We sincerely apologize for the confusion caused by the figure rendering issue in the paper, which was unexpected on our end. This appears to be a bug introduced in Microsoft Edge Stable version 136.0.3240.50 (released after the paper submission deadline). Currently, there are several workarounds: (1) disable the new PDF viewer in Edge, (2) use any other browser engine (e.g. Chrome) or standalone PDF reader. We are actively working on fixing the formatting to ensure full compatibility across different PDF viewers for the revision. Thank you for your understanding.
>
>
> >**W2: The main text describes diffusion defined on a continuous latent space, yet Figure 1 illustrates diffusion applied to Euclidean coordinates, which is confusing.**
>
> Sorry for the unclear description. Our model performs diffusion over a *joint* latent pair $Z_t = (H_t, X_t)$, where $H_t$ contains SE(3)-**invariant** features (e.g., sequence or chemical embeddings), and $X_t \in \mathbb{R}^{N \times 3}$ contains SE(3)-**equivariant** spatial coordinates. In the figure, each point cloud represents a block: its position corresponds to $X_t$, while its color encodes $H_t$. The current caption only mentions $Z$, without clearly distinguishing the roles of $H$ and $X$, which understandably led to confusion. We will revise the caption to clarify this distinction.
>
>
>
> >**W3: The model’s generalizability may be limited to targets that have strong binders represented in the retrieval database, potentially restricting applicability to novel targets.**
>
>
> Thank you for raising this insightful question! Our experiments indicate that even a single retrieved binder can provide meaningful guidance for generation. Furthermore, the cross-domain retrieval is exactly designed to enhance generalization by increasing the likelihood of discovering similar interaction patterns across different contexts.
>
> In rare cases where even no suitable retrieval data is available across all the databases and modalities, we propose constructing an artificial retrieval database using a protein design pipeline. As shown in the table below, we used Trop2 as the target—currently lacking any experimentally resolved binder structures—and evaluated two settings: (1) no retrieval and (2) retrieval using a synthetic database constructed with RFdiffusion.
>
> **Table S1: Ablation study on the retrieval stage**
> |Settings|DockQ|Rosetta $\Delta G$|Foldx $\Delta G$|
> |-|-|-|-|
> |（1）|0.549|-24.68|-6.54|
> |（2）|0.574|-25.74|-6.86|
>
>
> The results clearly show that retrieval using the artificial database yields significantly better performance. This suggests a practical solution for extremely novel cases where not a single structurally similar binders is available.
>
>
> >**Q1: Is there a measurable correlation between the binding affinity of retrieved binders and that of the generated binders?**
>
>
> Thank you for the insightful comment. We computed the Pearson correlation coefficient between the Rosetta dG scores of generated binders and those of the retrieved binders, which yielded a value of 0.6620—indicating a strong correlation. As a control, we also calculated the correlation between generated binders and randomly selected samples from the retrieval database, which resulted in a much lower value of 0.0536. These findings further support the effectiveness of our RAG framework in improving the binding affinity of generated samples.
>
>
> >**Q2: The baseline comparison may be biased, as the proposed model retrieves binders from multiple types while baselines are trained on a single binder type. Could the authors report performance in a more directly comparable setting, where the training data and binder types are matched?**
>
>
>
> Thank you for highlighting this point. First, we note that UniMoMo also involves multiple data types, making it a direct and fair comparison. Additionally, we performed an ablation study using single-domain data for both training and retrieval. The results are summarized in the table below:
>
> **Table S2: Ablation study on single-domain data**
> |Task and models|AAR (\%)|RMSD (Å)|$\Delta\Delta G$(kJ/mol)|IMP (\%)|ISM (\%)|
> |-|-|-|-|-|-|
> |UniMoMo(AbH3)|48.78|1.39|-5.781|63.33|65.46|
> |RADiAnce(AbH3)|51.31|1.109|-5.994|68.33|69.71|
> |UniMoMo(pep)|37.59|2.48|7.69|29.03|40.08|
> |RADiAnce(pep)|38.28|2.37|7.57|27.95|45.37|
>
> We observe that even when trained and retrieved within a single domain, RADiAnce still outperforms other methods under the same conditions. We will include the full ablation details in the supplementary material. We appreciate your thoughtful comments and suggestions!
>
> >**Q3: How do the model determined the number of residues during inference?**
>
> Thank you for the question, and we apologize for not detailing this in the main text due to space limitations. For benchmarking evaluations, we follow common conventions by using the sequence length as the reference for fair comparison[1]. For real-world applications on *de novo* design, our framework allows users to manually select the binding pocket and specify a desired sequence length range with flexibility.
>
> [1] Luo, Shitong, et al. "Antigen-specific antibody design and optimization with diffusion-based generative models for protein structures." Advances in Neural Information Processing Systems 35 (2022): 9754-9767.

---

> > ### Comment · Reviewer_J8pu · 2025-08-05
> >
> > Thank you for your detailed response and additional experiments, which address my concerns. I have one question about your RFdiffussion generated synthetic dataset. Are you generating using the binding site information? How is this dataset constructed?

---

> > > ### Author Response · Authors · 2025-08-06
> > >
> > > Thank you for your reply and follow-up questions! To build our synthetic dataset with RFdiffusion, we begin by using **prior binding site information** to guide the generation process. From the generated candidates, we evaluate binding quality based on predicted binding energies and select those samples demonstrating strong binding affinities to form our retrieval dataset. Finally, our model is used to generate new samples conditioned on the same binding site.
> > >
> > > This experiment simulates conditions in which the original retrieval database contains no similar interfaces, and the synthetic dataset provides a practical solution when retrieval data are lacking.

---

> > > ### Author Response · Authors · 2025-08-07
> > >
> > > Dear Reviewer J8pu,
> > >
> > > We are glad to hear that our previous responses have addressed your concerns. As the end of the discussion period is approaching, we are eager to hear your further replies on our response to your follow-up question on the RFdiffussion-generated synthetic dataset.
> > >
> > > We hope that these updates and clarifications fully resolve your concerns. If there are any remaining questions, we would greatly appreciate your additional feedback. We also kindly invite you to reconsider your evaluation based on our responses and new results, which we will appreciate a lot.
> > >
> > > Thank you once again for your thorough and constructive review, which is of great importance to our work!
> > >
> > > Best regards,
> > >
> > > The Authors

---

### Official Review · Reviewer_yozy · 2025-07-02

**Clarity:** 3
**Significance:** 3
**Originality:** 3
**Rating:** 4
**Confidence:** 4

**Summary:**

The paper presents a new method for generating protein binders using the retrieval-augmented generation (RAG) technique. A dataset includes pairs of binding sites and binders, which are encoded into a shared latent space. Contrastive learning is employed to align the representations of binding sites and binders. A generator, trained as a VAE model, takes latent space vectors and decodes binders conditioned on the binding site. Additionally, the top n binders are retrieved from a key-value database to perform a latent diffusion process that produces block types and their positions. The model is tested in two different setups. First, the retrieval capabilities are evaluated by querying antibodies and peptides. Then, the generative capabilities are assessed, demonstrating promising results both in terms of predicted binding strength and interactions at the binding site.

**Questions:**

1. What is a block type? Is it one of the standard amino acids? Can this method generate other amino acids?
2. Why is $\Delta\Delta G$ positive in Table 2? Do you believe that designing peptides is more challenging than designing antibodies?
3. What is the relationship between the number of retrieved examples and the model's performance? For example, is a single instance enough to significantly improve results?

**Ethical Concerns:**

["NO or VERY MINOR ethics concerns only"]

**Final Justification:**

All issues I raised were resolved in the rebuttal and during further discussion with the Authors. I believe this is a strong paper with valuable insights on using retrieval-augmented generation for molecular data. I recommend acceptance because the Authors properly addressed my concerns, and the inclusion of confidence intervals makes the results more reliable. All additional experiments, metrics, and related work discussed with the reviewers will need to be incorporated into the paper.

**Limitations:**

One limitation of the method is discussed in the “Conclusions and Limitations” section, which is its reliance on retrieval quality. Other limitations may include the small number of amino acids that can be produced, even though unconventional amino acids are sometimes preferred in real-world peptide design cases (see Question 1).

**Quality:**

3

**Strengths And Weaknesses:**

Strengths:
- The choice of the model architecture and the way retrieved examples are integrated appears well-justified.
- Multiple models for peptide design are included in the benchmarks, including UniMoMo, which is the same core model but without RAG.
- The paper is very clear, and the illustrations help make the method easier to understand. Especially Figure 2, which I find extremely helpful.
- The proposed latent RAG improves the quality of the generated protein binders, outperforming other methods.
- The proposition is original because there are only a few papers that employ RAG for molecule generation, and to the best of my knowledge, this is the first paper to apply RAG to protein design.

Weaknesses:
- Experiments should be repeated multiple times (at least three) to report error bars of the resulting performance. The checklist (point 7) states that error bars are reported, but in reality, they are not. If the authors claim that the results are stable and representative, this should be confirmed with some experiments, such as repeated training or evaluation runs.
- The related work should include a discussion about RAG in molecule generation. At least f-RAG should be covered [1], but during the review period, more notable papers appeared, including READ [2] and Rag2Mol [3].
- Multiple datasets are combined to form a training dataset for the proposed model. The methods used to clean the data and verify its quality should be described. If standard test sets are used for these datasets, are similar molecules removed from the training data across data sources?
- “Block type” should be defined.
- $\mathcal{Z}_x^t$ should also be defined. In the text, only $\mathcal{Z}_x$ is used before Section 3.3. In Figure 2, $\mathcal{Z}_t$ is used instead.

Minor comments:
- On page 5, the sentence “The layer is inserted between the self-attention layer and the feed-forward layer in the original EPT” is repeated.
- A typo on page 6:: “accurate retrieval is is essential”
- A typo on page 8: “we randomly mask one CDR loops”

[1] Lee, Seul, et al. "Molecule generation with fragment retrieval augmentation." Advances in Neural Information Processing Systems 37 (2024): 132463-132490.

[2] Xu, Dong, et al. "Reimagining Target-Aware Molecular Generation through Retrieval-Enhanced Aligned Diffusion." arXiv preprint arXiv:2506.14488 (2025).

[3] Zhang, Peidong, et al. "Rag2Mol: Structure-based drug design based on Retrieval Augmented Generation." Briefings in Bioinformatics 26.3 (2025): bbaf265.

---

> ### Author Rebuttal · Authors · 2025-07-31
>
> Thanks for your appreciation and the positive comments!
>
> >**W1: Experiments should be repeated multiple times (at least three) to report error bars of the resulting performance. The checklist (point 7) states that error bars are reported, but in reality, they are not. If the authors claim that the results are stable and representative, this should be confirmed with some experiments, such as repeated training or evaluation runs.**
>
> Thank you for raising this important point. We apologize for the confusion caused by omitting explicit error bars in the original submission. Each evaluation in our study aggregates results over several thousand samples, leading to highly stable metrics. To substantiate this claim, we report the standard deviation of all key metrics as the table below. The resulting deviations are consistently small, confirming the robustness and reproducibility of our evaluation.
>
> **Table S1: Performance metrics with standard deviations**
> |Model|AAR (\%)|RMSD (Å)|$\Delta\Delta G$ (kJ/mol)|IMP (\%)|ISM (\%)|
> |-|-|-|-|-|-|
> |HCDR3|54.66±0.0026|0.9443±0.0156|-6.236±0.5862|71.67±0.0096|71.64±0.0085|
> |Peptide|39.42±0.0014|2.29±0.0043|1.963±0.2302|41.94±0.0108|52.15±0.0007|
>
> >**W2: The related work should include a discussion about RAG in molecule generation. At least f-RAG should be covered [1], but during the review period, more notable papers appeared, including READ [2] and Rag2Mol [3].**
>
>
> Thank you for drawing our attention to the recent lines of work on retrieval-augmented generation for molecular design.
> Retrieval-augmented generation (RAG) has emerged as a powerful paradigm for targeted molecular design, with **f‑RAG** introducing a dynamic vocabulary of molecular fragments to improve small molecule generation through fragment-level retrieval [1],[2]. Structure-based RAG methods such as **IRDiff** incorporate high-affinity ligands as retrieval references to steer diffusion-based generation in protein-specific 3D molecular design[3], and **READ** further integrates pocket-matched scaffold embeddings via SE(3)-equivariant diffusion to ensure valid, affinity-optimized ligand generation[1]. The **ChemRAG‑Bench** benchmark demonstrates that RAG strategies yield an average relative improvement of about 17% over direct generation across diverse chemistry tasks[4]. However, none of these approaches address the challenges of protein binder co-design, which requires simultaneous modeling of sequence and structure, nor provide a unified retrieval framework across modalities such as peptides and antibodies. Our method, **RADiAnce**, addresses this gap by learning a contrastive latent space that aligns interaction interfaces across diverse binder modalities, and uses a retrieval-augmented latent diffusion model to enable cross-domain motif retrieval guiding joint sequence–structure binder design.
>
> [1] Xu D, Yang Z, Wong K, et al. Reimagining Target-Aware Molecular Generation through Retrieval-Enhanced Aligned Diffusion[J]. arXiv preprint arXiv:2506.14488, 2025.
>
> [2] Lee, Seul, et al. "Molecule generation with fragment retrieval augmentation." Advances in Neural Information Processing Systems 37 (2024): 132463-132490.
>
> [3] Huang, Zhilin, et al. "Interaction-based retrieval-augmented diffusion models for protein-specific 3d molecule generation." Forty-first International Conference on Machine Learning. 2024.
>
> [4] Zhong, Xianrui, et al. "Benchmarking retrieval-augmented generation for chemistry." arXiv preprint arXiv:2505.07671 (2025).
>
> In the revised version we will expand the *Related Work* section to cover these approaches and clarify how our framework differs and advances beyond them.
>
>
> >**W3: Multiple datasets are combined to form a training dataset for the proposed model. The methods used to clean the data and verify its quality should be described. If standard test sets are used for these datasets, are similar molecules removed from the training data across data sources?**
> >
>
> Thanks for your concern about the data splits!
> We follow established practice by adopting RABD as the antibody benchmark and LNR as the peptide‑design benchmark.
> To prevent data leakage, we clustered all sequences at 40 % sequence identity and assigned the train, validation, and test splits by clusters. Consequently, no sequence in test or validation shares a cluster with any training sequence.
> We further verified the split by computing (i) the sequence‑identity and (ii) TM‑Scores between every test‑set structure and the training set. Both metrics confirmed low similarity, indicating that our split is stringent and leak‑free.
>
> **Table S2: Data split verification metrics**
> |Testset|TMscore with Trainset|SeqId with Trainset|
> |-|-|-|
> |RABD|0.2421±0.053|0.1272±0.021|
> |LNR|0.2503±0.069|0.0983±0.017|
>
> >**W4: “Block type” should be defined.**
> >**Q1: What is a block type? Is it one of the standard amino acids? Can this method generate other amino acids?**
>
> We apologize for the confusion on the block type. Following UniMoMo, we treat each standard amino acid as one block, and use principal subgraph decomposition algorithm to decompose other entities (e.g. non-standard amino acids) into blocks of frequently occurring chemical motifs. The resulting vocabulary $V$ thus extends beyond standard amino acids and also includes principal subgraphs derived from non-standard components.
> We will incorporate these revisions in the final version of the paper. Thank you for your valuable feedback.
>
> >**W5: Zxt should also be defined. In the text, Zx only is used before Section 3.3. In Figure 2, Zt is used instead.**
>
> We apologize for the confusion. In our notation, $Z_x$ denotes the equivariant features of the binder side, whereas $Z_t$ in Fig. 2 represents the latent embedding of the entire graph (binder + binding site), including both the invariant feature and the equivariant feature. We will update the figure and provide precise symbol definitions to make this distinction clear.
>
>
>
> >**Q2: Why is $\Delta\Delta G$ positive in Table 2? Do you believe that designing peptides is more challenging than designing antibodies?**
>
> Thank you for the insightful comment. We clarify that the reported $\Delta\Delta G$ refers to the difference in binding free energy between the sampled structure and the reference structure, where a negative value indicates that the sampled structure exhibits stronger binding. Notably, peptide binders possess fewer conformational constraints compared to antibodies, leading to greater flexibility and a significantly larger search space[1][2][3], which makes identifying effective binders more challenging. However, we would like to emphasize that despite these difficulties, our model still achieves state-of-the-art performance compared to existing baselines.
>
> [1] London, Nir, Dana Movshovitz-Attias, and Ora Schueler-Furman. "The structural basis of peptide-protein binding strategies." Structure 18.2 (2010): 188-199.
>
> [2] Lee, Jessica H., et al. "Structural features of antibody-peptide recognition." Frontiers in Immunology 13 (2022): 910367.
>
> [3] Makabe, Koki. "Molecular basis of flexible peptide recognition by an antibody." The journal of biochemistry 167.4 (2020): 343-345.
> >**Q3: What is the relationship between the number of retrieved examples and the model's performance? For example, is a single instance enough to significantly improve results?**
>
> Thank you for raising the valuable question! Indeed, the number of retrieved samples does have an impact, and even a single retrieved sample can lead to noticeable performance improvements, as shown in the tables below.
>
> For HCDR3:
>
> **Table S3: Ablation study on the number of retrieved samples for HCDR3**
> |Retrieved Samples|AAR (%)|RMSD (Å)|Codesign Diversity|
> |-|-|-|-|
> |0|51.55|0.9525|0.0524|
> |1|53.26|0.9431|0.0404|
> |10|54.66|0.9443|0.0484|
> |20|54.63|0.9682|0.0485|
> |40|54.06|0.9653|0.0494|
>
>
> And for Peptide:
>
> **Table S4: Ablation study on the number of retrieved samples for Peptide**
> |Retrieved Samples|AAR (%)|RMSD (Å)|Codesign Diversity|
> |-|-|-|-|
> |0|37.94|2.38|0.5042|
> |1|39.03|2.38|0.4846|
> |10|39.12|2.31|0.5129|
> |20|39.42|2.29|0.5227|
> |40|38.16|2.28|0.5273|
>
>
> We could find that increasing the number of retrieved samples generally leads to slightly better results, albeit with a modest increase in computational cost. However, overly increasing the number might include samples with lower similarity score, introducing noise into the generation and thus harming the performance. Notably, using no retrieval sample performs significantly worse than using any retrieval.
>
> An additional observation is that the diversity of generated outputs also varies with the number of retrieved samples. Specifically, both the no-retrieval setting and the setting with many retrieved samples exhibit higher diversity, while retrieving only a single sample results in the lowest diversity. This aligns with our expectation, as limited retrieval can overly constrain the generative space.
>
> These findings suggest that practitioners may choose an appropriate top‑N retrieval size based on the trade-off between performance, diversity, and efficiency.

---

> > ### Comment · Reviewer_yozy · 2025-08-04
> >
> > Thank you for your response! I still have two questions that need clarification:
> > 1. I do not understand how error bars are calculated. Are they just standard deviations of errors across testing data points? If so, these are not proper error bars, which should be based on repeated experiments with different model initializations (model variability under different random seeds) or different testing datasets (data variability, such as using cross-validation).
> > 2. The differences between using 1 and 10 nearest neighbors in Tables S3 and S4 seem quite small. Have you also examined how the distance to these nearest examples influences prediction improvement? Are predictions worse when the closest latent binders are dissimilar to the query?

---

> ### Author Response · Authors · 2025-08-04
>
> Thank you very much for raising these important points. We sincerely appreciate your careful consideration, and below we address your concerns in detail.
>
> >**R1: I do not understand how error bars are calculated. Are they just standard deviations of errors across testing data points? If so, these are not proper error bars, which should be based on repeated experiments with different model initializations (model variability under different random seeds) or different testing datasets (data variability, such as using cross-validation).**
>
> Thank you for raising this and for your careful attention to the evaluation protocol. We apologize for not stating this clearly earlier. In the original submission we reported only means; in the rebuttal, the error bars denote **standard deviations across independent runs with different random seeds** (we re-ran the entire inference and evaluation pipeline three times). They are **not** computed across test data points. The per-example dispersion across test points is indeed much larger, but it reflects heterogeneity of individual samples rather than run-to-run uncertainty; with a large test set, the aggregate metrics are stable and the variability across seeds is small.
>
> > **R2: The differences between using 1 and 10 nearest neighbors in Tables S3 and S4 seem quite small. Have you also examined how the distance to these nearest examples influences prediction improvement? Are predictions worse when the closest latent binders are dissimilar to the query?**
>
> Thank you for this insightful question. Yes, we find that retrieval quality, not just quantity, is the key factor. When the retrieved neighbors are dissimilar to the query, performance degrades. In our ablations, increasing the number of retrieved samples tends to admit more low-similarity examples (the average similarity drops as k grows), and metrics worsen accordingly. Conversely, when the retrieved set maintains higher average similarity, performance improves.
>
> **Table S3: Ablation study on the number of retrieved samples for HCDR3**
>
> | Retrieved | Similarity |   AAR (%) |   RMSD (Å) | Codesign Diversity |
> | --------: | ---------: | --------: | ---------: | -----------------: |
> |  0 | N/A |     51.55 |     0.9525 |  **0.0524** |
> |  1 | 57.62±5.25 |     53.26 | **0.9431** |0.0404 |
> | 10 | 54.65±4.90 | **54.66** |     0.9443 |0.0484 |
> | 20 | 50.43±4.78 |     54.63 |     0.9682 |0.0485 |
> | 40 | 46.04±4.67 |     54.06 |     0.9653 |0.0494 |
>
> **Table S4: Ablation study on the number of retrieved samples for HCDR3**
>
> | Retrieved | Similarity |   AAR (%) | RMSD (Å) | Codesign Diversity |
> | --------: | ---------: | --------: | -------: | -----------------: |
> |  0 | N/A |     37.94 |     2.38 |0.5042 |
> |  1 | 55.56±4.49 |     39.03 |     2.38 |0.4846 |
> | 10 | 51.67±3.60 |     39.12 |     2.31 |0.5129 |
> | 20 | 50.24±3.30 | **39.42** |     2.29 |0.5227 |
> | 40 | 47.68±3.12 |     38.16 | **2.28** |  **0.5273** |
>
> To address this, we replace fixed top-N retrieval with an **adaptive similarity-cutoff**: only neighbors above a threshold are used, so the method automatically uses more neighbors when many are relevant and fewer when they are not.We set the similarity cutoff to 51.0 for HCDR3 and 53.0 for peptides, chosen to approximate the average similarity between the ground-truth binder and the binding-site embeddings.
>
> We compare (1) the fixed top-N baseline with (2) this adaptive cutoff approach:
>
> **Table S5: Performance with adaptive retrieval for HCDR3**
>
> | Settings     | AAR (%) |   RMSD (Å) | Codesign Diversity | ΔΔG (kJ/mol) | FoldX ΔG (kJ/mol) | ISM (%) |DockQ |
> | ------------ | ------: | ---------: | -----------------: | -----------: | ----------------: | ------: | ---------: |
> | (1) |   54.63 |     0.9682 |0.0485 |-6.236 |    -8.4250 |   71.64 |     0.9582 |
> | (2) |   **54.71** | **0.9147** |  **0.0523** |   **-8.938** |   **-8.6180** |   71.61 | **0.9604** |
>
> **Table S6: Performance with adaptive retrieval for Peptide**
>
> | Settings     | AAR (%) | RMSD (Å) | Codesign Diversity | ΔΔG (kJ/mol) | FoldX ΔG (kJ/mol) |   ISM (%) |DockQ |
> | ------------ | ------: | -------: | -----------------: | -----------: | ----------------: | --------: | ---------: |
> | (1) |   39.42 |     2.29 |0.5227 | 1.963 |-7.99 |     52.15 |     0.7698 |
> | (2) |   39.22 | **2.27** |  **0.5400** |    **1.427** |-7.97 | **52.89** | **0.7762** |
>
> Across both tasks, the adaptive approach improves multiple metrics, especially diversity and binding energies. These results indicate that filtering by similarity preserves the benefits of larger retrieval pools while avoiding noise from less relevant neighbors.
>
> In summary, predictions are indeed worse when the nearest latent binders are dissimilar; enforcing a similarity cutoff mitigates this by filtering out low-relevance neighbors. We believe this provides a general and practical solution that can be tuned to task requirements, and we will include the method and implementation details in the revision.

---

> ### Author Response · Authors · 2025-08-07
>
> Dear Reviewer yozy,
>
> As the discussion period is ending soon, we would be grateful for any additional feedback you might have on our manuscript and rebuttal. Thank you again for your detailed review and constructive questions. Your comments have already helped us improve the work substantially.
>
> Below we briefly recap how we have addressed each of your remaining points:
>
> | Point          | Our response|
> | - | - |
> | **\[W1,R1]** | Re-ran the entire inference and evaluation pipeline **three times** with different random seeds; we now report means ± standard deviations, confirming robustness.                                                        |
> | **\[Q3,R2]**     | Analysed the joint effect of average similarity and retrieved-set size on performance, and proposed an **adaptive retrieval strategy** that maintains high quality while adjusting k automatically.                     |
>
> We hope these additions and clarifications fully address your concerns. If any point remains unclear, we would sincerely appreciate your further comments. We would also be grateful if you could reconsider your evaluation in light of the new results.
>
> Thank you again for your time and valuable feedback.
>
> Best regards,
>
> The Authors

---

> > ### Comment · Reviewer_yozy · 2025-08-08
> >
> > Thank you for your response! This addresses all my remaining concerns. The adaptive similarity cutoff idea is particularly interesting. I am glad to see that this resulted in improved model performance.

---

> > > ### Author Response · Authors · 2025-08-09
> > >
> > > We are pleased to learn that your concerns have been fully addressed. We greatly value your careful and detailed review, which has significantly contributed to improving the quality of our work. We will incorporate the points discussed into the revised manuscript, and we sincerely appreciate your constructive and thoughtful feedback.

---

### Author Response · Authors · 2025-08-09
**Summary of Rebuttal and Discussion Outcomes**

We sincerely thank the Area Chair for organizing and facilitating the rebuttal and discussion, and for the thoughtful guidance throughout the process. We also extend our gratitude to all reviewers for their constructive feedback, which motivated additional experiments, deeper analyses, and clearer presentation of our work.

In this work, we propose **RADiAnce**, a retrieval-augmented diffusion framework for **sequence and structure codesign** of protein binders across modalities (antibodies and peptides). Our key contributions are:

* **Contrastive latent space for cross-domain interface retrieval,** enabling a unified similarity metric across different binder types.
* **First application of retrieval-augmented generation (RAG) to binder codesign** in a shared latent space with the retriever, where retrieved interface embeddings guide generation via cross-attention and residual conditioning.
* **Extensive benchmarking and analysis.**

In the rebuttal and discussion phase, we addressed the reviewers’ concerns and provided additional experiments to further clarify our contributions. Key points include:

* **Strengthened evaluation** by reporting standard deviations for all metrics, and by conducting a detailed analysis of correlations between train and test data. Through stricter evaluations, we further demonstrated the effectiveness of our work.
* **Demonstrated generalization** by employing a synthetic retrieval database for novel targets.
* **Explored the relationship** between the number of retrieved samples, their average similarity to the query, and overall model performance, and ultimately introduced an **adaptive similarity cutoff** to balance retrieval quantity and quality, achieving the best results.

These new results have substantially enriched the paper, especially in terms of **rigor in data splitting** and the **practicality and flexibility of the model** in real world applications. The additional findings addressed the reviewers’ concerns, **Lic4**, **KBot**, and **yozy** expressed positive views on the new results and explanations, and **J8pu** acknowledged that most of his concerns were resolved while raising an additional question about synthetic retrieval database. We provided a detailed explanation to this question, though no further feedback was received before the discussion deadline.

Overall, the rebuttal and discussion phase was productive, yielding valuable insights and substantial improvements to the paper. We deeply appreciate the reviewers’ careful evaluation and the Area Chair’s efforts in guiding the process.

---

### Decision · Program_Chairs · 2025-09-17

**Decision:**

Accept (poster)

**Comment:**

The paper presents RADIANCE: a RAG-based method for generating protein binders. Briefly, the method aligns pairs of binders in a latent space through contrastive learning. A generator, trained as a VAE model, takes latent space vectors and decodes binders conditioned on the binding site. Additional information about (top-n) related systems is integrated in latent diffusion process to generate binders.
The paper is methodologically interesting and evaluation is done carefully. Especially after the review, rebuttal and discussion, several issues concerning for instance, data-leakage, was quite convincingly resolved.